# Genome-wide analyses of variance in blood cell phenotypes provide new insights into complex trait biology and prediction

Ruidong Xiang [1,2,3,4,5] ✉, Chief Ben-Eghan[2,6,7,8,9], Yang Liu[1,2,6,7,8,9], David Roberts[10,11], Scott Ritchie [1,2,6,7,8,9], Samuel A. Lambert [1,2,6,7,8,9], Yu Xu [2,6,7,8,9], Fumihiko Takeuchi [1,12] & Michael Inouye [1,2,6,7,8,9] ✉

Blood cell phenotypes are routinely tested in healthcare to inform clinical decisions. Genetic variants influencing mean blood cell phenotypes have been used to understand disease aetiology and improve prediction; however, additional information may be captured by genetic effects on observed variance. Here, we mapped variance quantitative trait loci (vQTL), i.e. genetic loci associated with trait variance, for 29 blood cell phenotypes from the UK Biobank (N ~ 408,111). We discovered 176 independent blood cell vQTLs, of which 147 were not found by additive QTL mapping. vQTLs displayed on average 1.8-fold stronger negative selection than additive QTL, highlighting that selection acts to reduce extreme blood cell phenotypes. Variance polygenic scores (vPGSs) were constructed to stratify individuals in the INTERVAL cohort (N ~ 40,466), where the genetically most variable individuals had increased conventional PGS accuracy (by ~19%) relative to the genetically least variable individuals. Genetic prediction of blood cell traits improved by ~10% on average combining PGS with vPGS. Using Mendelian randomisation and vPGS association analyses, we found that alcohol consumption significantly increased blood cell trait variances highlighting the utility of blood cell vQTLs and vPGSs to provide novel insight into phenotype aetiology as well as improve prediction.

The complete blood count is amongst the most routinely ordered clinical laboratory tests performed globally[1]. Blood cells play crucial roles in a variety of biological processes, such as oxygen transport, iron homoeostasis, and pathogen clearance[2–4], and serve as key biological conduits for interactions between an individual and their environment. The genetic architecture of blood cell traits has been recently elucidated by genome-wide association studies (GWAS)[5,6] and, consistent with their well-known role in disease and clinical testing, blood cell

[1]Cambridge Baker Systems Genomics Initiative, Baker Heart and Diabetes Institute, Melbourne, VIC, Australia. [2]Cambridge Baker Systems Genomics Initiative, Department of Public Health and Primary Care, University of Cambridge, Cambridge, UK. [3]Agriculture Victoria, AgriBio, Centre for AgriBioscience, Bundoora, VIC 3083, Australia. [4]The School of Applied Systems Biology, La Trobe University, Melbourne, VIC 3086, Australia. [5]Baker Department of Cardiometabolic Health, The University of Melbourne, Melbourne, VIC 3010, Australia. [6]British Heart Foundation Cardiovascular Epidemiology Unit, Department of Public Health and Primary Care, University of Cambridge, Cambridge, UK. [7]Victor Phillip Dahdaleh Heart and Lung Research Institute, University of Cambridge, Cambridge, UK. [8]Health Data Research UK Cambridge, Wellcome Genome Campus and University of Cambridge, Cambridge, UK. [9]British Heart Foundation Centre of Research Excellence, University of Cambridge, Cambridge, UK. [10]National Institute for Health Research Blood and Transplant Research Unit in Donor Health and Genomics, University of Cambridge, Cambridge, UK. [11]NHS Blood and Transplant–Oxford Centre, John Radcliffe Hospital and Radcliffe Department of Medicine, University of Oxford, John Radcliffe Hospital, Oxford, UK. [12]Department of Bioinformatics, National Center for Global Health and Medicine, Tokyo, Japan. ✉e-mail: ruidong.xiang@agriculture.vic.gov.au; mi336@cam.ac.uk

traits are both highly heritable and have been genetically linked to many diseases, including cardiovascular diseases[7], mental disorders[8] and autoimmune diseases[9].

Despite the success of GWAS, our understanding of the genetic architecture of complex traits has been limited by a focus on mean trait values and how these change with respect to genotype. The genetics of trait variance, how individual measurements deviate from the mean trait value across genotypes, is far less studied. It has long been known that trait variance, e.g. for gene expression[10,11] and metabolic rate[12], plays a role in an organism's fitness and phenotypic penetrance. Theories support the existence of selection on trait variance to improve fitness[13,14]. However, there are limited observations of selection on clinically significant traits. Variance quantitative trait loci (vQTLs) have been identified for human body composition traits, such as body mass index (BMI)[15,16], and for cardiometabolic biomarkers[17]. vQTLs have also been linked to gene-by-environment interactions (GxE) or gene-by-gene interactions (GxG)[15–18]. vQTL studies of blood cell traits are currently lacking, despite their central role in biological processes and ubiquity in clinical testing.

Polygenic scores (PGS) are being intensively studied in various ways to determine their utility in clinical practice[19–21]. PGS for blood cell traits, in particular, are both highly predictive and show sex- and age-specific interactions[6,7]. How to treat trait variance and vQTLs with respect to phenotype prediction is relatively unexplored. A variance PGS (vPGS) to predict the trait variance may be estimated from the effect sizes obtained from a genome-wide vQTL analysis. In theory, a PGS is different from a vPGS, where the former may be used to stratify individuals based on the inherited trait level while the latter stratifies individuals based on the inherited deviation of individuals from the population mean. It is known that the accuracy of a PGS varies across individuals as a function of the genetic distance from the reference population[22]. As a vPGS may represent the outcome of GxE[16] or GxG due to the nature of vQTLs[15], examining a PGS alongside vPGS may reveal individual variability in PGS accuracy that can be accommodated.

Here, we conduct genome-wide vQTL analysis for 29 blood cell traits in individuals of European ancestries in UK Biobank[6,7] and the INTERVAL cohort[23]. We compared the discovered vQTL with conventional QTL and analysed vPGS with conventional PGS in the prediction of blood cell traits. We found novel vQTL which displayed strong selection to reduce blood cell trait variances. Finally, we demonstrate the use of vPGS in stratifying individuals, resulting in differing PGS performance, and then show that PGS performance within vPGS strata is associated with lifestyle factors.

## Results

### Genome-wide discovery and annotation of vQTLs in the UK Biobank

We performed GWAS of variance in 29 blood cell traits from the UKB[17,18] (Average sample size = 402,142, Supplementary Data 1). The processing of phenotypes and genotypes followed previously established protocols with stringent quality control and normalisation procedures[5–7]. Levene's test[24], a robust test for equality of variances[24] across (genotype) groups, as implemented in OSCA[25], was used to map vQTLs for each of the 29 blood cell traits. We also compared OSCA and an alternative method, the deviation regression model (DRM)[26], by checking their summed polygenic effects of vQTLs across INTERVAL individuals, which exhibited a correlation of 0.904, suggesting high consistency (Supplementary Fig. 1). The inflation factors and lambda GC were assessed using LD Score regression (LDSC)[27], a GWAS-summary data-based genetic analysis method. Across the 29 traits, the average lambda GC and LDSC intercepts were 1.03 and 1.007, respectively (Supplementary Data 2), indicating negligible inflation. At a study-wide significance level of $p < 4.6 \times 10^{-9}$ and with clumping $r^2 < 0.01$, we identified 176 independent vQTLs (Fig. 1a, Supplementary Data 3, "Methods").

Basophil cell count (baso) and basophil percentage of white cells (baso_p) yielded the largest number of independent vQTLs ($N = 27$ and 23, respectively), whereas high light scatter reticulocyte count (hlr) did not have any study-wide significant vQTLs (Supplementary Data 4). Most vQTL were associated with the variance of only one or two traits and many of these traits were correlated (Supplementary Fig. 2 and Supplementary Data 3). By counting the number of blood cell traits associated, the most pleiotropic lead vQTL was located in gene *HBM* (haemoglobin subunit mu) and was associated with the variance of four traits (red blood cell count, mean corpuscular volume, mean corpuscular haemoglobin and mean corpuscular haemoglobin concentration, Supplementary Data 3). The second-most pleiotropic lead vQTL related to long intergenic non-coding RNA *LINCO2768* was associated with 3 traits [monocyte percentage of white cells (mono_p), baso and baso_p, Fig. 1b]. To account for the phenotypic correlations, the pleiotropy of trait variance was further assessed using HOPS[28], which found that 495 SNPs (out of 71,216 input SNPs) showed significant pleiotropy (Supplementary Data 5). In this analysis, the most significant pleiotropic locus was *LINCO2768* (Supplementary Data 5).

vQTLs were largely distinct from additive QTLs. Of 176 lead vQTLs, 147 were not detected as additive QTLs by Vuckovic et al.[6], the largest GWAS to date of blood cell traits. vQTLs had an average $r^2$ of 0.33 (SD = 0.12) with the lead additive QTLs from Vuckovic et al.[6] (Supplementary Fig. 3). We repeated the OSCA[25] analysis fitting the trait level as a covariate, i.e. effects of vQTL conditioned on the trait level. The correlation of the effects of these vQTLs between the original and conditional analysis was 0.99 (Supplementary Fig. 4), consistent with vQTL effects being independent of those for mean trait level.

Across 29 traits, the magnitude of the genetic correlation between trait variance and trait level, as estimated by LDSC[27], was on average 0.328 (SD = 0.24) (Fig. 1c, Supplementary Data 6) and the genetic correlation between trait variance and value was not significant for 21 out of 29 traits after adjusting for multi-testing (FDR corrected, same below). Notably, red cell distribution width (rdw) and neutrophil percentage of white cells (neut_p) had significant negative genetic correlations between their levels and variances after adjustment for multiple testing, indicating genetic control of trait variance so it is reduced at high levels of rdw or neut_p. Rdw is itself a measure of variation of red cell widths, and high rdw is an indicator of iron or other nutrient deficiencies. Therefore, our results suggest a potential simultaneous genetic stabilisation when rdw is genetically high. Similarly, high neut_p is an indicator of microbial or inflammatory stress, thus a negative genetic correlation between the level and variance suggests a stabilisation at genetically high neut_p levels.

With many known trait-associated alleles under negative selection[29], we also assessed the extent to which QTLs for trait variability were under selection. We used Bayes(S)[29], a Bayesian method to detect the relationship between SNP effect size and minor allele frequency, to compare the selection coefficient (S) between vQTLs and additive QTLs across 29 blood cell traits (Fig. 1d). We found that, on average S was 1.8 times stronger on trait variance (−0.82, SD = 0.07) than trait level (−0.45, SD = 0.05) (Fig. 1d, Supplementary Data 7). These results show a much stronger negative selection on blood cell trait variance than on trait level. The correlation of S between trait variance and level was positive but not significant (r = 0.14, $p = 0.46$, Supplementary Fig. 5). While it can be difficult to differentiate between negative and stabilising selection, our results indicate negative selection is acting on both vQTLs and additive QTLs (somewhat more so on the former than the latter) to remove extreme blood cell phenotypes from the population.

We applied FUMA[25], a platform to annotate, prioritize, visualize and interpret GWAS results to the lead vQTLs for each trait (Supplementary Data 8–9) and performed a trait enrichment analysis with GWAS Catalogue[23]. We found multiple significant overlaps between vQTL and additive QTL related to alcohol consumption. Significant

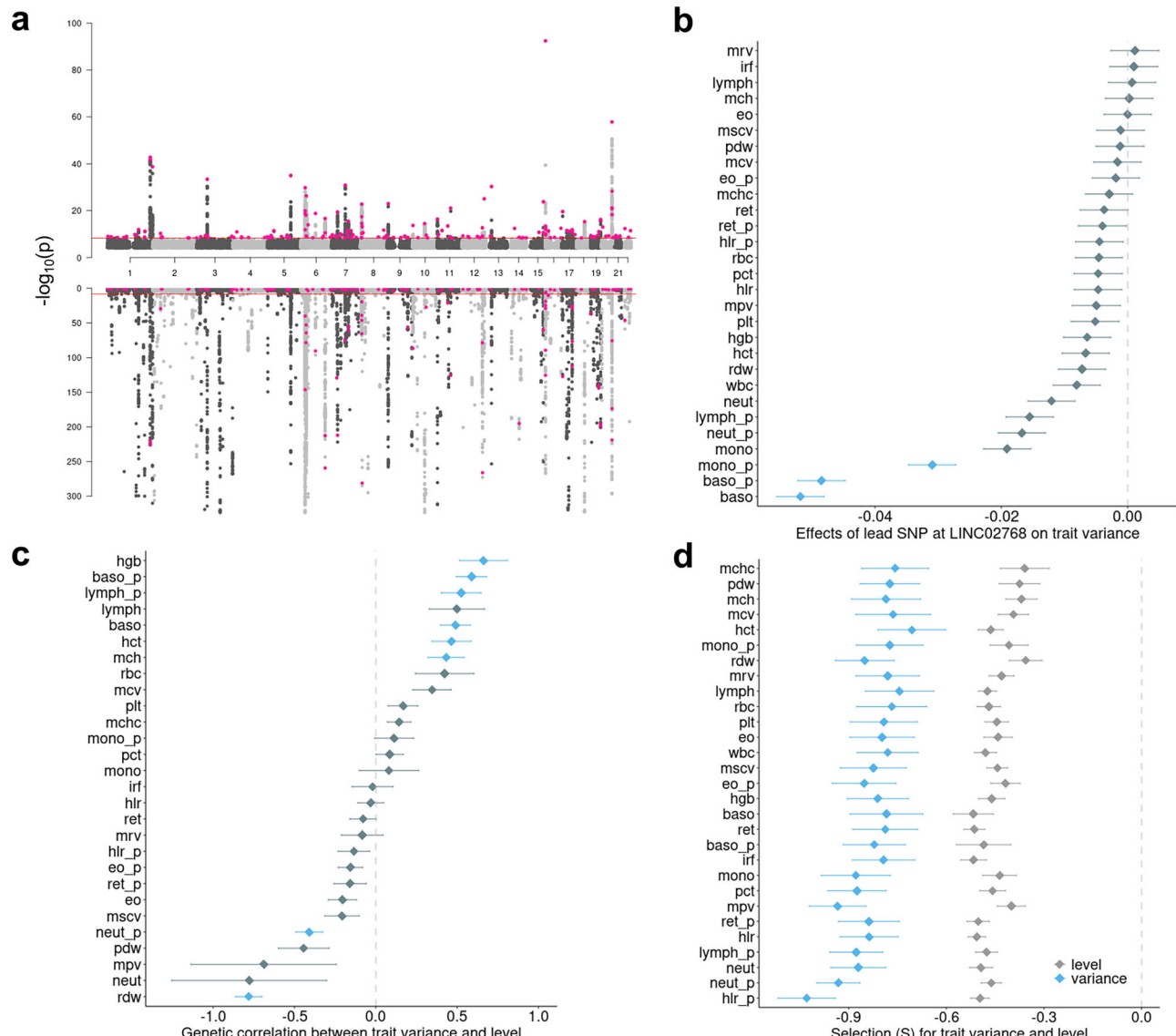

**Fig. 1 | vQTLs for 29 blood cell traits and their comparison with additive QTLs.**
**a** Miami plot showing the best (smallest nominal $p$ value, Levene's test, see methods) vQTL across 29 blood cell traits (top plot) and the corresponding best additive QTLs (bottom plot). Red dots are genome-wide significant independent vQTLs.
**b** Example of pleiotropic effects of the C allele of rs10803164 for the long noncoding RNA *LINC02768* on blood cell trait variance. Blue indicates the effect on trait variance had $p < 4.6 \times 10^{-9}$ (nominal study-wide GWAS significance, Levene's test,

see "Methods" and Data Availability). **c** Genetic correlation (LDSC) between blood cell trait variance and trait level. Blue indicates the correlation had multi-testing adjusted $p < 0.05$ (Supplementary Data 6). **d** Selection coefficient estimated by BayesS[29] for trait variance and level. All analyses used UK Biobank data with sample size $\sim N \sim 408{,}111$. In panels (**b**–**d**), data are presented as mean values ± SEM. Full names of blood cell traits can be found in Supplementary Data 1.

vQTLs (rs191673261 in LD with lead vQTL rs572454376) for platelet crit (pct) were located proximal to *ALDH2*, a well-known gene contributing to alcohol consumption[30] (Fig. 2a). Lead vQTLs were also significantly enriched for GxE interactions ("Methods") with age, sex, BMI, smoking status, and alcohol consumption (Supplementary Fig. 6; study FDR < $5.5 \times 10^{-5}$, Supplementary Data 10), with alcohol consumption having the largest number of significant effects of interactions with lead vQTLs on blood cell traits. The genetic correlation between alcohol consumption and blood cell trait variance estimated using LDSC had an average magnitude of 0.1 (SD = 0.08) (Supplementary Data 11).

We subsequently performed Summary-data-based Mendelian Randomisation (GSMR)[31] between GWAS of alcohol consumption (as exposure, obtained from Cole et al.[32]) and variances of blood cell traits (as outcome). Of note, Mendelian randomisation is a technique that uses SNPs as instrumental variables to infer potential causal

associations between phenotypes[33]. Sensitivity analyses to the assumptions underlying GSMR were performed using MR-PRESSO[34] and MR-weighted median[35] (Supplementary Data 11). We did not find statistically significant causal links between alcohol consumption and pct. However, at multi-testing adjusted $p < 0.05$ level, increased alcohol consumption was genetically predicted to increase variance in mean corpuscular volume (mcv) and mean sphered corpuscular volume (mscv) (Fig. 2b–d). At nominal significance ($p < 0.05$ for each of the three MR methods), increased alcohol consumption was genetically predicted to increase variance in red blood cell count (rbc) and neutrophil percentage of white cells (neut_p) (Fig. 2b). The positive effects of alcohol consumption on neutrophil count (neut) were significant in GSMR (nominal p = 0.014) and MR-PRESSO (nominal $p = 0.008$), but insignificant (nominal $p = 0.1$) in MR-weighted median. Overall, our results support alcohol consumption as affecting particular blood cell trait variances. There was a significant correlation

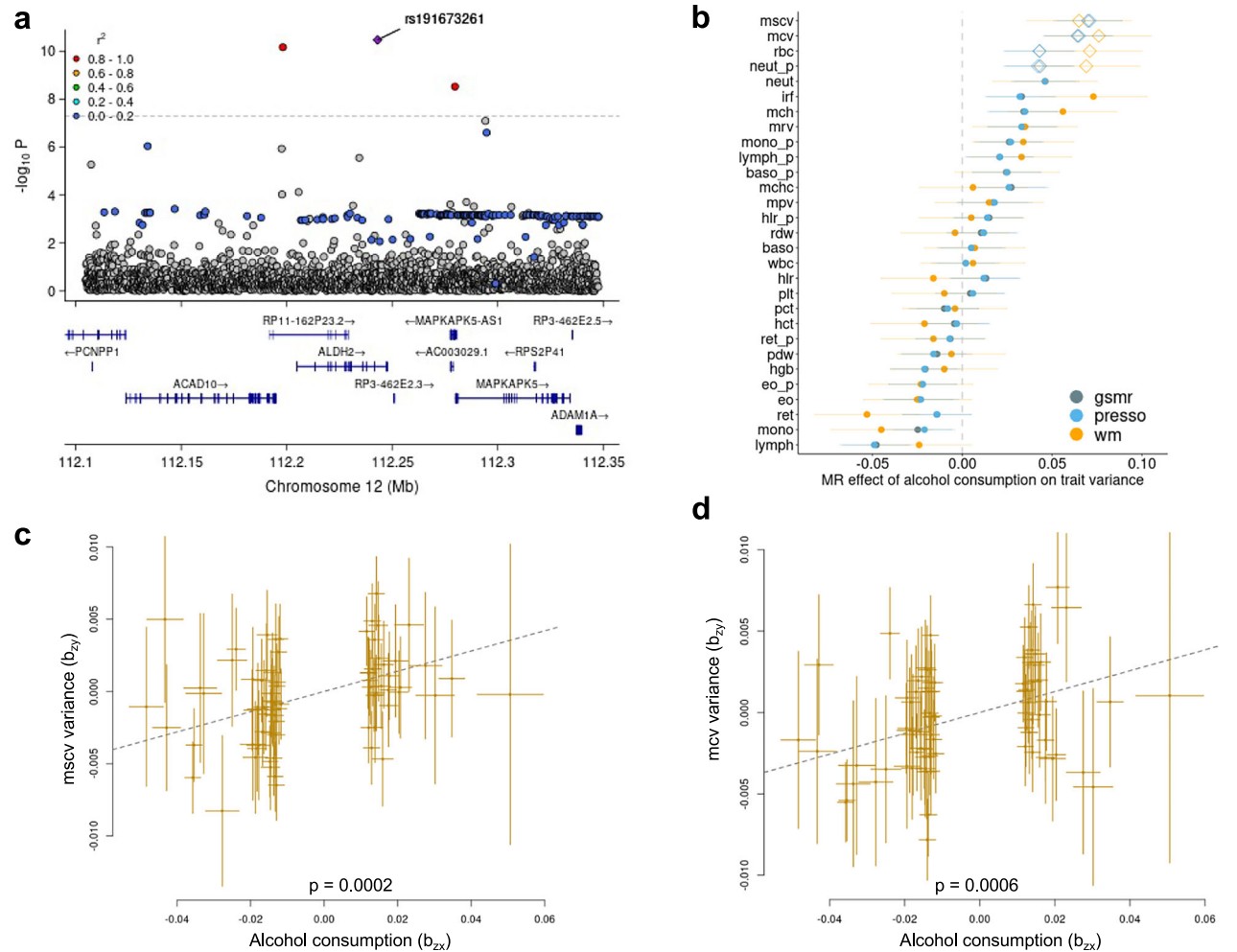

**Fig. 2 | Relationships between alcohol consumption and blood cell trait variances. a** LocuzZoom plot of variance QTL mapping for platelet crit (pct) variance at ALDH2 gene (Levene's test, see "Methods"); (**b**) Mendelian randomization (MR) of alcohol consumption on variance of blood cell traits using GSMR[31], MR-PRESSO (presso)[34] and weighted-median (wm)[35]. Diamonds: significant in 3 methods. **c** Effects of MR of alcohol consumption on variance of corpuscular haemoglobin concentration (mscv); **d** Effects of MR of alcohol consumption on variance of

corpuscular volume variance (mcv). Dashed fitted lines indicate the coefficient of Mendelian Randomisation ($b_{xy} = 0.07$, $se_{xy} = 0.019$ for mscv and $b_{xy} = 0.064$, $se_{xy} = 0.0188$ for mcv). In panels (**c**, **d**), multi-test adjusted $p$ values are shown. All analyses used UK Biobank data with sample size -N - 408,111. In panels (**b**–**d**), data are presented as mean values ± SEM. Full names of blood cell traits can be found in Supplementary Data 1.

between genetic correlation coefficients and the effects of Mendelian randomisation ($r = 0.5$, $p = 0.006$, Supplementary Fig. 7). Together, our results support the genetic link between alcohol consumption and the variance of blood cell trait variances.

FUMA-enabled ANNOVAR[24] was used to study the enrichment of vQTLs in different functional annotation classes. We found that vQTLs for mean sphered corpuscular volume (mscv), reticulocyte count (ret) and reticulocyte fraction of red cells (ret_p) were significantly enriched in exonic variants related to protein-coding functions (Supplementary Fig. 8a). However, vQTLs for many other traits were enriched in regulatory regions. For example, vQTLs for mean corpuscular haemoglobin concentration, red blood cell count and haemoglobin concentration (hgb) were enriched for upstream gene regulatory sites. vQTLs for eosinophil count (eo), mean corpuscular haemoglobin and mean corpuscular volume were enriched for downstream regulatory sites of genes. vQTLs for platelet distribution width (pdw) and basophil percentage of white cells (baso_p) were enriched for UTR-3' sites (Supplementary Fig. 8a). We used pathway enrichment analyses within FUMA to further investigate whether vQTLs were enriched for gene regulation, finding that vQTLs for mean corpuscular haemoglobin were enriched for many epigenetic

regulatory mechanisms including DNA methylation and histone modifications (Supplementary Fig. 8b).

## Polygenic scores of blood cell trait variance

Polygenic scores, a predictor of an individual's genetic predisposition for a given phenotype[21], are conventionally constructed for differences in trait level. Using the vQTL results from the UK Biobank and applying the same concept of estimating PGSs, we constructed polygenic scores for blood cell trait variance (vPGS) using PRSICE[36] and the INTERVAL study as an external validation cohort (Supplementary Data 1, "Methods"). For conventional PGS we utilised those from Xu et al.[7]. Across 27 blood cell traits available in INTERVAL, there was nearly zero Pearson correlation between vPGS and PGS (mean 0.00028, range [-0.018, 0.023]; Supplementary Fig. 9), consistent with PGS for trait variance being independent from those for mean trait levels.

A potential use of vPGS is to stratify a population by trait variance, thus identifying subgroups where predictive models may have increased performance. For each trait, we stratified individuals into the top and bottom 5% of vPGS. As vPGS were trained to estimate SNP effects on trait variance, individuals with lower or higher vPGS were expected to display less or more variation around the trait mean,

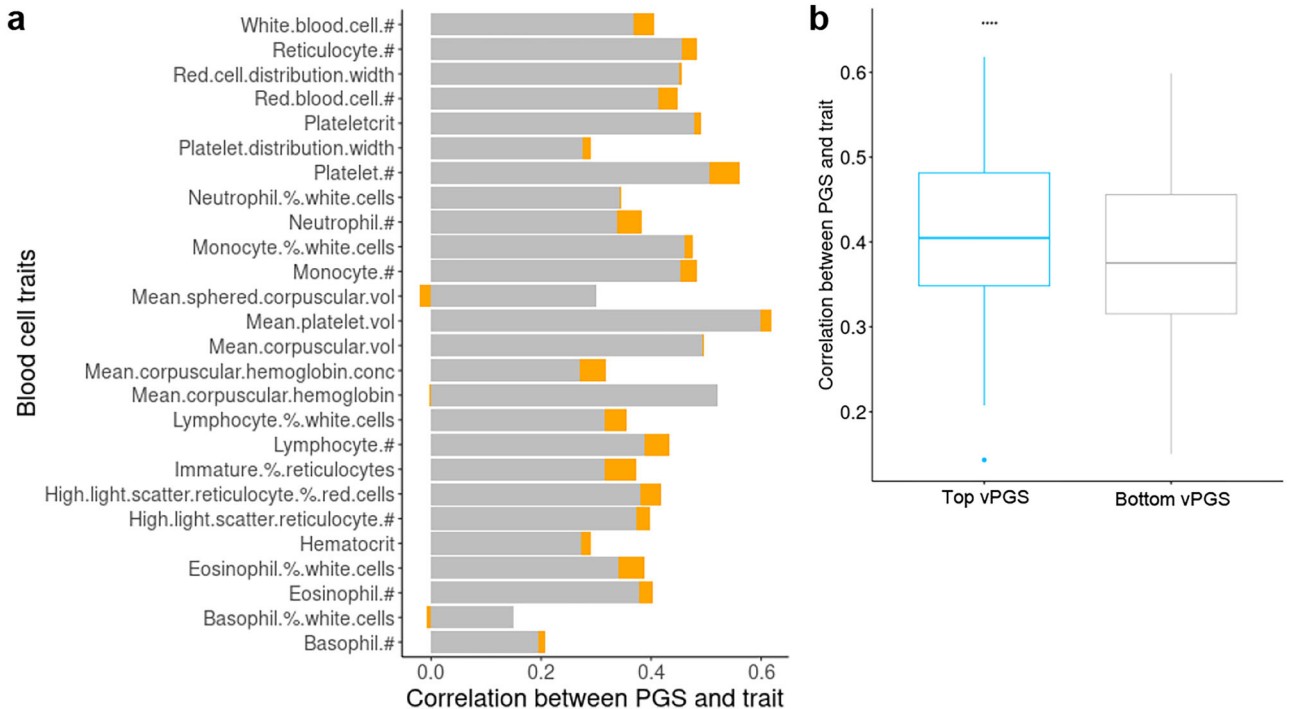

**Fig. 3 | The variation in the accuracy of PGSs for 27 blood cell traits (Pearson correlation) between the top and bottom vPGS groups. a** Accuracy of PGS in the top vPGS group (more variable group, grey colour) and the difference (orange) of PGS between the top vPGS group (most variable group) and the bottom vPGS group (less variable group). #: count; % percentage; vol: volume; conc: concentration. **b** Difference of accuracy of PGS between the top and bottom vPGS groups across 27 blood cell traits. ****$p$ (2-side test) <0.0001. For each box, the minimum is the lowest point, the maximum is the highest point, whiskers are maxima 1.5 times of interquartile range, the bottom bound, middle line and top bound of the box are the 25th percentile, median and the 75th percentile, respectively.

respectively. We then compared the correlation of PGSs for each trait between these more (high-vPGS) or less variable (low-vPGS) groups. Across the 27 blood cell traits, we found the high-vPGS group (top 5% of vPGS) had a significantly higher PGS-trait correlation than the low-vPGS group (bottom 5% vPGS) (Fig. 3). Across all traits, the mean relative difference in PGS-trait correlation (Pearson) between the high-vPGS and low-vPGS groups was +6.5% [-7%, 18%] (Fig. 3), with a mean difference of +6.6% [−9%, 19%] for spearman correlation (Supplementary Fig. 10). We expanded this analysis using 10 vPGS bins and compared this result with the stratification using 10 PGS bins (Supplementary Fig. 11). While the results regarding vPGS stratification remained largely the same as described above (Supplementary Fig. 11b), individuals within the top and bottom PGS bins had the highest PGS accuracy compared to those within the middle bins of PGS (Supplementary Fig. 11a).

Next, we analysed the effects of interaction between PGS and vPGS for each trait. We found that 6 out of 27 blood cell traits displayed statistically significant ($p < 0.05$) effects of interaction between PGS and blood cell trait level (Fig. 4a), suggesting that associations between PGS and blood cell trait level can depend on vPGS (Fig. 4b, c). For seven traits (eo, rbc, plt, neut, mcv, baso and lymph, Supplementary Data 12), the main effects of their corresponding vPGS were also significant. As expected, the effects of vPGS were much smaller than PGS on trait levels, as the PGSs are directly estimated from trait levels.

Next, for all INTERVAL individuals, we examined whether adding vPGS to PGS increased the prediction of blood cell trait level. For each blood cell trait, we estimated the difference in the variance explained ($R^2$) between PGS models with or without vPGS (Fig. 5, "Methods"). Across all 27 traits, the mean $R^2$ increase was +1.8% (range [0%, 5%]) and 9 traits showed a statistically significant[37] increase in $R^2$ (Fig. 5, "Methods"). We further tested whether multi-trait vPGSs also increase prediction power[38], and found that adding multi-trait vPGSs to PGS

increased $R^2$ by a mean of +3.5% (range [0%, 10%]) and the increase was statistically significant in 16 traits (Fig. 5).

### Lifestyle effects on blood cell trait variance
To investigate why some individuals had highly variable blood cell trait levels we assessed the effects of alcohol consumption along with other lifestyle variables such as smoking behaviour, age, BMI and sex. We first identified distinct groups of individuals with high or low trait variance in INTERVAL. For the high variability trait group, we identified individuals who were in the top 5% of vPGS for at least 4 blood cell traits and, for the low variability trait group, with individuals in the bottom 5% of vPGS for at least 4 traits ("Methods", Fig. 6). Our analysis found that those in the high variability trait group were more likely to be current or previous consumers of alcohol (Fig. 6a). Further, we applied this analysis to mcv, neut_p and rbc, finding significant putative causal effects of alcohol consumption in GSMR analyses (Fig. 2a, mscv not available in INTERVAL). Consistent with the results from GSMR, individuals with high variability in mcv, neut_p and rbc were more likely to be alcohol consumers (Fig. 6b). These results are also supported by additional analyses testing for association between observed phenotypic variances in blood cell traits and alcohol consumption in the UKB (Supplementary Fig. 12). Together, our results support the hypothesis that alcohol consumption increases variation in blood cell traits.

### Discussion
The analysis of vQTL and vPGS may yield new insights into locus and GxE discovery as well as the use of human genetics for patient stratification, as shown by previous studies[15,17]. Our study explored vQTL analysis in 29 blood cell traits in the UK Biobank, where the majority (84%) of vQTLs did not overlap with and were largely independent of genetic variants identified in conventional GWAS of trait mean. We

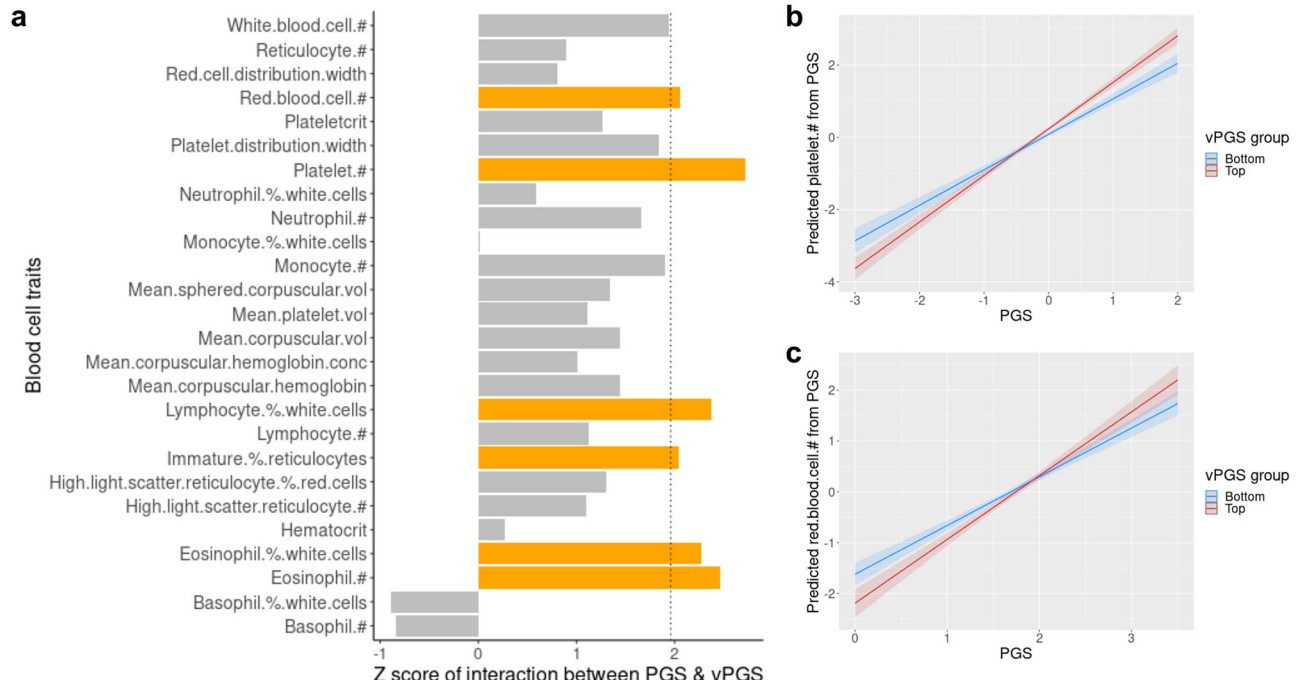

**Fig. 4 | Effects of interaction between PGS and vPGS on blood cell traits.**
**a** Effects of interaction across 27 traits in INTERVAL (Supplementary Data 12). The vertical dashed line indicates the z-score test statistic value = 1.96 which equals nominal $p$ value = 0.05 and bars with z-score value > 1.96 (nominal 2-sided $p < 0.05$) are in orange colour. #: count; % percentage; vol: volume; conc: concentration. **b**, **c** Examples of visualised effects of interaction for eosinophil percentage of white cells (eo_p) and neutrophil count (neut). Data are presented as mean values ± SEM.

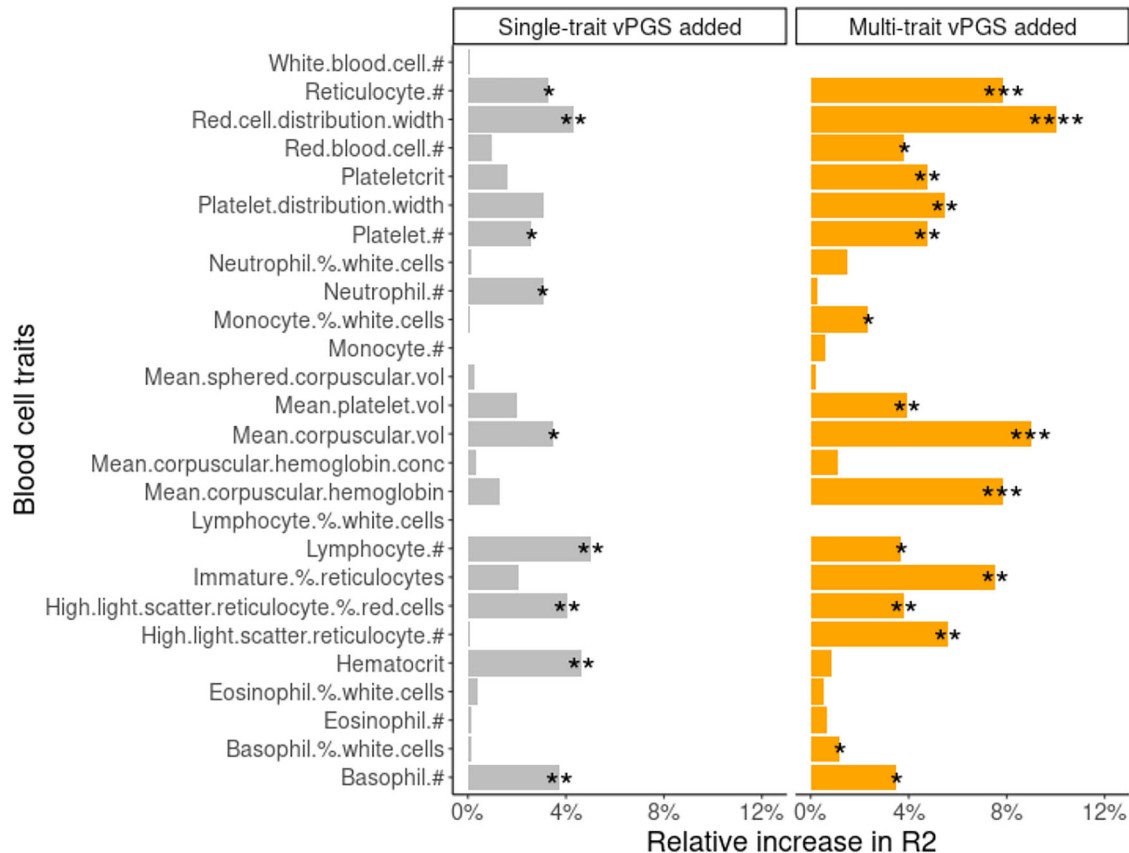

**Fig. 5 | The difference in the variance explained ($R^2$) between PGS models with or without vPGS.** Each bar represents the relative increase in $R^2$ (model goodness of fit) for the blood cell trait when the PGS model added vPGS. In the left panel, the single-trait vPGS was added to PGS. In the right panel, multi-trait vPGS was added to PGS. #: count; % percentage; vol: volume; conc: concentration. *$p < 0.05$; **$p < 0.01$; ***$p < 0.001$ and ****$p < 0.0001$. nominal 2-sided $p$ values were estimated by comparing models with and without vPGS using r2redux[37].

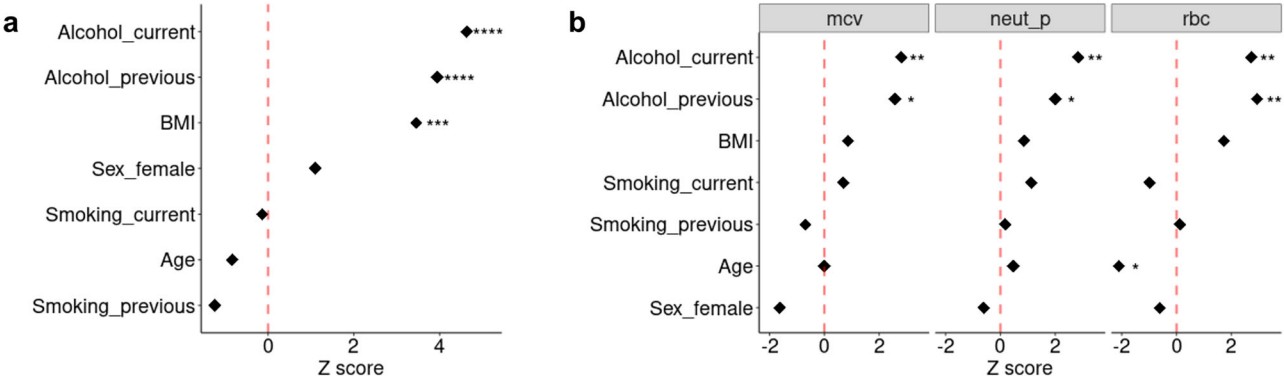

**Fig. 6 | Association between BMI, age, alcohol drinking and smoking and individuals to be genetically variable across blood cell traits in INTERVAL. a** An overall Z score test estimate across 27 blood cell traits. **b** Z score test estimates for mean corpuscular volume (mcv), neutrophil percentage of white cells (neut_p) and red blood cell count (rbc) which were significant Mendelian Randomisation analyses. Z score = beta (effects) / se (standard error). * (nominal 2-sided): $p < 0.05$; **: $p < 0.01$; ***: $p < 0.001$ and ****: $p < 0.0001$.

investigated the functional annotation, pathway-level associations and selection of vQTLs. The potential utility of using vQTLs to construct vPGS and using the latter to stratify the population into groups of trait variance was demonstrated. Finally, our analysis also showed trait variance to be related to non-genetic factors, finding that alcohol consumption had a putatively causal effect on increasing blood cell trait variances.

Both blood cell trait variances and levels displayed significant negative selection. Stabilising selection of human traits has been reported[14]. However, to our knowledge, negative selection on blood cell trait variance, particularly its strength relative to that on trait level, has not yet been identified. Evolutionary theories show that stabilizing selection will reduce phenotypic variations to maintain population fitness[13,39–42]. Our results are in line with these theories, although we caution not to overinterpret with respect to the magnitude of negative selection. However, the highly significant negative selection of blood cell trait variances suggests that extreme blood cell levels and morphologies (some of which may be indicative of disease) have not generally been favoured. Selection to reduce phenotypic variances implies stabilising selection for blood cell trait levels, which has been reported for other human traits[14]. For example, we observed stabilising selection for neutrophil percentage of white blood cells alongside negative selection for neutrophil levels. Neutrophils are innate immune cells that act as first responders against infection by releasing cytotoxic antimicrobial peptides; damaging proximal tissue at their site of activity[43]. Increased neutrophil abundance and activity are associated with myriad chronic inflammatory conditions and are predictive of long-term risk of cardiovascular risk and mortality[44,45]. Importantly, hyperactivity of the innate immune response from severe infection events (such as COVID-19) can lead to "cytokine storms"; causing extensive tissue damage and rapidly leading to organ failure and death[46]. Therefore, selection to reduce both extremely high levels of neutrophils and their variance may reflect evolutionary mechanisms acting on immune systems to improve survival.

Many vQTLs tagged loci implicated in GxG, GxE and under epigenetic regulation, consistent with previous studies of vQTLs[18,47]. We found blood cell vQTLs tagged genes related to diet. Previous GWAS of diet identified loci related to blood lipids[48] and glycated haemoglobin[49] but not to blood cell traits analysed here; however, others have reported that alcohol intake increases mean corpuscular volume independent of the genetic contribution to the level of mean corpuscular volume[50]. The association between alcohol and macrocytosis is well-established[51,52]. In our study, alcohol consumption-related loci significantly overlapped with vQTLs for platelet count, the function of which can be significantly affected by alcohol drinking[53].

Stratification by vPGS was shown to identify groups with significantly different PGS prediction accuracy, indicating that some groups are intrinsically harder to predict by PGSs than others. Illustrating this point, our analysis found multiple significant interactions between PGS and vPGS. This implies that the effects of PGS on the phenotype can depend on vPGS, which suggests that the non-additive and GxE components related to PGSs could impact prediction accuracy. These findings are consistent with previous observations[54,55] and may be important for PGS translation. However, to be clear, our observation of the interactions between PGS and vPGS is purely statistical. Future research integrating further molecular data in observational or experimental settings may refine our understanding of these interactions. Nevertheless, we speculate that vPGS could add to PGS to increase genomic prediction performance for those patients at risk. In addition, we identified a list of vQTLs significantly enriched in genomic and epigenomic regulations (Supplementary Data 1), highlighting genes which may be useful for future research on therapeutic targets.

Our results also showed that alcohol consumption and, to some extent increased BMI, were significant contributors to increased genetic variability in blood cell traits. Previously reports have found that blood cell traits can be significantly influenced by alcohol intake[56] and BMI[57]. However, to our knowledge, this is the first study to report lifestyle risk factors contributing to genetically predicted variation in blood cell traits. In the Mendelian Randomisation analysis, we have chosen alcohol consumption as the exposure as established evidence supports the adverse effects of alcohol drinking on blood cell morphologies[56,58,59], likely due to mediation by inflammation and immune responses[60]. We used GSMR as a discovery tool for Mendelian Randomisation analyses and verification with MR-PRESSO and weighted median. While GSMR and MR-PRESSO both correct for pleiotropic confounders, there could exist other confounders not accounted for in the current study, thus we caution that this is evidence for, but not proof of, a potentially causal interpretation of the effects of alcohol consumption on blood cell trait variances.

Our study has several limitations. For example, this study uses existing methods to understand vQTLs and vPGS, which are still being explored. Our vPGS was computed using parameters implemented in PRSice-2[36] and reported by Miao et al.[16], which may provide a conservative estimate of the effects of vPGSs as PGSs developed by pruning+thresholding are usually underpowered[61,62]. Future studies in developing and comparing different methods, and testing them in cohorts and ancestries beyond the UK Biobank and INTERVAL will deepen our understanding of vQTLs and vPGSs. While our study provides proof-of-concept evidence for vPGS to be informative on top of

conventional PGSs, the current results are still several steps away from clinical application. Therefore, any potential future role in clinical practice with vPGS will depend on myriad factors, including the infrastructure to deploy even conventional PGSs, quantification of clinical utility, and assessments of demographic transferability. To provide this proof-of-concept, our study was also limited to only European ancestries. However, the challenge of transferability of genetic signals and PGS across ancestries is of high importance and much further research in diverse human populations with paired genomic and blood cell trait data is necessary. Such research should initially focus on multi-ancestry vQTL mapping, combining single-ancestry vQTL mapping results using sophisticated meta-analysis methods, and extend to the latest polygenic score approaches that prioritise ancestry transferability, such as PRS-CSx[63]. While we employed multiple methods to infer putative causal relationships between alcohol consumption and blood cell variances, to confirm these relationships there will be a need for further triangulation, e.g. via experimental or trial-based evidence.

In conclusion, our study provides an in-depth analysis of human genetic effects on the variance of blood cell traits, including the discovery of loci and strong negative selection, improved genomic prediction and stratification, and identification of GxE, such as the effect of alcohol consumption is genetically linked to blood cell variances. vPGSs may provide a generalisable approach to incorporate individual differences to improve trait and disease risk prediction. This study demonstrates that there is substantive human biology and potential clinical utility in studying trait variances alongside conventional studies of trait means.

## Methods

### Study cohorts and methods

**UK biobank.** The UK Biobank[64,65] (https://www.ukbiobank.ac.uk/) is a cohort including 500,000 individuals living in the UK who were recruited between 2006 and 2010, aged between 40 and 69 years at recruitment. Our research complies with all relevant ethical regulations. Ethics approval was obtained from the North West Multi-Centre Research Ethics Committee. The current analysis was approved under UK Biobank Project 30418. The participants with the measurements of the 29 blood cell traits and who were identified as European ancestry based on their genetic component analysis were included in our study. The detailed sample sizes used for vQTL detection were shown in Supplementary Data 1.

**INTERVAL Study.** INTERVAL[23] (https://www.intervalstudy.org.uk/) is a randomised trial of 50,000 healthy blood donors, aged 18 years or older at recruitment. The participants with measurements of the 27 considered blood cell traits were included in our study. The detailed sample sizes were shown in Supplementary Data 1. All participants have given informed consent and this study was approved by the National Research Ethics Service (11/EE/0538). All participants have given informed consent and this study was approved by the National Research Ethics Service (11/EE/0538).

**Data quality control.** For trait levels of 29 blood cell traits in the UK Biobank and matching 27 traits in the INTERVAL, we adopted previously established protocols for quality controls[5–7] to adjust technical and other confounders and the first 10 genetic principal components. For trait levels, adjusted technical variables include the time between venepuncture and full blood cell analysis, seasonal effects, centre of sample collection, the time-dependent drift of equipment, and systematic differences in equipment; other adjusted variables included sex, age, diet, smoking and alcohol consumption. The rationale for such adjustments was detailed in Astle et al. [5]. Briefly, after adjustment for age, sex, BMI and variables measuring smoking habits and alcohol consumption covariates still explained >= 0.5%

of variance blood cell traits. Therefore, all relevant environmental variables were included in the adjustments. We used the Kolmogorov-Smirnov Test to check the normality of phenotypes where the null hypothesis is the data comes from a normal distribution. The smallest $p$ value was 0.82 (Supplementary Data 13) so all traits are expected to be normally distributed after quality control done previously. Quality control and imputation of the genotype data have been described previously[5,65], which filtered the samples to the European ancestry only.

**vQTL analysis.** Genome-wide analysis of vQTL used Levene's test. As detailed in refs. 11,15, the test statistic of Levene's test is:

$$\frac{(n-k)}{(k-1)} \frac{\sum_{i=1}^{k} n_i (z_{i.} - z_{..})^2}{\sum_{i=1}^{k} \sum_{j=1}^{n_i} (z_{ij} - z_{i.})^2} \tag{1}$$

where $n$ is the total sample size, $k$ is the number of groups (k = 3 in vQTL analysis), $n_i$ is the sample size of the $i$th group (one of three genotypes), $z_{ij}$ is the absolute difference between the phenotype value in sample $j$ from genotype and the median value in genotype $i$, $z_{i.}$ is the average $z$ value in genotype $i$, and $z_{..}$ is the average $z$ value across all samples. OSCA-implemented Levene's test also provides beta and se estimates based on $p$ value and minor allele frequency[15] and the beta estimates were used to construct vPGSs described later. Levene's test relies on the assumption of normal distribution[15,24] which was for all blood cell traits as described above.

We estimated the study-wise significance for vQTL as $4.6 \times 10^{-9} = 5 \times 10^{-8} / 10.2$ where 10.2 is the effective number of traits analysed in the study. The effective number of traits is estimated using

$$\frac{(\sum_{k=1}^{p} \lambda_k)^2}{\sum_{k=1}^{p} \lambda_k^2} \tag{2}$$

where $\lambda_1 ... \lambda_p$ is principal component variances or the ordered eigenvalues[15,17]. To identify lead vQTL with relative independence, we first used LD-clumping[66] using a $p$ value threshold of $4.6 \times 10^{-9}$, $r^2 < 0.01$ and window size of 5000 kb (the same parameter used by ref.[15]). The LD analysis between vQTL and lead QTL reported by Vuckovic et al. [6] used plink 1.9 with the function of --ld. Second, as there are between trait correlations, i.e., blood cell phenotypes correlate with each other, a novel vQTL was defined as follows: (1) was a lead vQTL from the above described clumping analysis, (2) clumped lead vQTL did not have $p$ value < $4.6 \times 10^{-9}$ for any blood cell trait levels in Vuckovic et al. and (3) was not in strong LD ($r^2 < 0.8$) with reported lead QTL for any trait in Vuckovic et al. For vQTL mapping results of each trait, we used LDSC[27] to estimate lambda-GC and intercept to check inflation. We also used FUMA[67] to annotate significant vQTL for each trait with default settings. Results from FUMA functions of SNP2GENE and GENE2FUNC were presented in the results.

The targeted analysis of GxE used identified lead vQTLs and tested their effects of interaction with each one of the environmental factors of age, alcohol consumption, BMI, sex and smoking on blood cell traits. The formula of interaction analysis was

$$y = \mu + \beta_g X_g + \beta_E X_E + \beta_{gE} X_g X_E + e \tag{3}$$

where $\beta$ was the fixed effects, $X_g$ were the genotype of SNP and $X_E$ was the environmental factor. Blood cell phenotypes were adjusted by the top 10 genetic PCs for non-related UKB-EUR participants, using --king-cutoff 0.0884 (to prune out first and second-degree relatives), age and sex; e.g., when testing for genotype-sex interactions, the phenotypes were adjusted for PCs and age; for genotype-age interactions, the phenotypes were adjusted for the PCs and sex. To test the significance of the overall association study of GxE, we employed the study false discovery rate estimated as the number of associations that were

significant to the number expected by chance[68,69]:

$$FDR_{study} = \frac{P(1 - \frac{A}{T})}{\frac{A}{T}(1 - P)} \quad (4)$$

where $P$ was the $p$ value threshold hold, $A$ was the number of significant associations and $T$ was the total number of associations tested. $FDR_{study} < 0.05$ is interpreted as a significant association study. For this analysis, only vQTLs with MAF $> = 0.001$ were considered.

To explore the potential causal relationships between alcohol consumption and blood cell trait variances, we used GSMR[31] to discover the putative causal relationships and used MR-PRESSO[34] and weighted-mean implemented in MendelianRandomisation[35] as validation. A key confounder of Mendelian randomisation is pleiotropy where variants can be naturally associated with multiple traits[70]. Employing multiple methods that account for pleiotropy is a common strategy to reduce false positives. The GWAS summary data for alcohol consumption was obtained from Cole et al. [32]. Default settings for nominated software were used and SNPs with $p$ value $< 5e-8$ and $r^2 < 0.05$ were used in the analysis. Significant results were defined as the multi-testing adjusted p value from GSMR $< 0.05$ and the nominal significance was defined as the Mendelian Randomisation had raw $p$ value $< 0.05$ in all 3 methods.

**Analysis of vPGS and PGS.** PGS trained using the elastic net from Xu 2022 et al. [7] was used. For training vPGS, we followed the protocol described by Miao et al. [16] reported successful implementation of vPGS for BMI using PRSICE[36], we used the same procedure described by Miao et al. to construct vPGS in the INTERVAL using PRSICE, i.e., –clump-p1 1 –clump-p2 1 –clump-r$^2$ 0.1 and –clump-kb 1000. When vPGS was computed for each trait, they were used to rank INTERVAL individuals where the top and bottom 5% of individuals were stratified. As vPGS was trained based on SNP effects on phenotypic variance, i.e., the extent to which the individual measurement deviates from the mean, vPGS was expected to genetically predict such variation of individuals for the corresponding trait. Therefore, individuals ranked in the top 5% of vPGS were called the genetically more variable group and individuals ranked in the bottom 5% of vPGS were called the genetically less variable group. Then, for each trait, within the more variable and less variable groups, we estimated the PGS accuracy, i.e., the correlation between PGS and the corresponding trait. We then compared the PGS accuracy between the more variable and less variable groups for each trait and the relative increase was calculated as

$$\frac{r_{less\ variable} - r_{more\ variable}}{r_{more\ variable}} \quad (5)$$

where $r_{less\ variable}$ is the PGS accuracy in the less variable group defined by vPGS and $r_{more\ variable}$ is the PGS accuracy in the more variable group defined by vPGS. The choice of the 5% top/bottom grouping is arbitrary, although the choice of 1% top/bottom would result in a very small sample size in each group. The results for the top/bottom 10% are consistent with the choice of the 5% top/bottom are shown in Supplementary Fig. 11, 13.

The effects of interaction between PGS and vPGS on the corresponding trait in INTERVAL were tested on corrected blood cell traits (described above). As the traits were already corrected for covariates, only the main effects and interaction of PGS and vPGS were fitted for each blood cell trait in the lm() function in R:

$$y = PGS + vPGS + PGS * vPGS \quad (6)$$

where y was each of the blood cell trait. The effects of interaction on specific traits (e.g., eo_p and neut) were visualised using the function of plot_model in the R package sjPlot (version 2.8.15).

To evaluate if adding vPGS improves PGS model predictability, we tested two sets of vPGS, where one set is the original single-trait vPGSs for 27 traits computed by PRSICE, and the other set is estimated using the multi-trait BLUP (SMTpred[38]) combining information from single-trait vPGSs. Following the instructions from https://github.com/uqrmaie1/smtpred, we used the LDSC[27] wrapper (ldsc_wrapper.py) with default options in SMRpred to estimate the genetic parameters for each trait which are required inputs by the multi-trait BLUP. Then, the script smtpred.py was used by default options with the estimated genetic parameters to combine single-trait vPGSs to construct multi-trait vPGSs. Then, we used r2redux[37] to quantify the difference in variance explained ($R^2$) between PGS models with and without vPGS. As described by Momin et al. [37], r2redux can powerfully detect $R^2$ differences between models for the out-of-sample genomic prediction which is suitable to our case where the PGS and vPGS models were trained in the UK Biobank and predicted into INTERVAL. We followed the instructions provided by (https://github.com/mommy003/r2redux) to compare the $R^2$ of models with vPGS and without PGS using the nested method and obtained p values testing the significance of the increase in $R^2$ when adding vPGS. The relative increase in $R^2$ was expressed as the absolute difference in $R^2$ divided by the heritability estimated using LDSC[27]. LDSC was also used to estimate the inflation factors, lambda GC and genetic correlation using default parameters. For estimating genetic correlations between alcohol consumption and blood cell genetic variances, the GWAS summary of alcohol consumption used Cole et al.[32].

To characterise the individuals that were identified as genetically variable across traits, we first counted the number of times (out of 27 blood cell traits) an individual was ranked in the top 5% by PGS for each trait. We also counted the number of times an individual was ranked in the bottom 5% by PGS for each trait. We then identified 2,465 individuals who always ranked in the top 5% vPGS, and 2,362 individuals who always ranked in the bottom 5% vPGS across multiple blood traits. Individuals in the top group were ranked in the top 5% vPGS for 4 to 17 traits with a mean of 5 and individuals in the bottom group were ranked in the bottom 5% vPGS for 4 to 23 traits with a mean of 9. Then, the top group was labelled as 1 and the bottom group was labelled as 0 and this 0/1 vector was analysed as a binary outcome for a logistic regression analysis against lifestyle factors:

$$y = age + sex + BMI + smoking\_status + drinking_{status} \quad (7)$$

where the average age is 46.1 (SD = 14.3) and the average BMI is 26.2 (SD = 4.6); for sex, there are 2,419 women; for smoking status, there are 2728 people never smoked, 378 current smokers, 1634 previous smokers and 87 with no answers; for alcohol drinking status, there are 118 who never drunk, 4178 current drinkers, 323 previous drinkers and 208 with no answers. The logistic regression used the function glm() in R and for sex the male was set to the reference level, for smoking the level of never smoked was set to the reference and for drinking the level of never drunk was set to the reference. We also tested the effects of alcohol intake which showed consistent results with less significance [most_days ($N = 251$), one_to_three_monthly ($N = 725$), one_two_weekly ($N = 1630$), three_five_weekly ($N = 1062$), and special_occasions (reference, $N = 600$), Supplementary Fig. 14]. The analysis with drinking status was also applied to individual blood cell traits of mean corpuscular volume (mcv), neutrophil percentage of white cells (neut_p) and red blood cell count (rbc) which were significant in Mendelian Randomisation analyses.

**Reporting summary**
Further information on research design is available in the Nature Portfolio Reporting Summary linked to this article.

## Data availability

Full summary statistics of vQTL mapping generated from this study are available via the GWAS Catalogue (https://www.ebi.ac.uk/gwas/) under the accession numbers GCST90565679-GCST90565707. Variance polygenic scores are available at the PGS Catalogue (https://www.pgscatalog.org/) under the accession number PGP000723 and scores PGS005172-PGS005197. The UK Biobank data are available through the UK Biobank subject to approval from the UK Biobank access committee. See https://www.ukbiobank.ac.uk/enable-your-research/apply-for-access for further details. INTERVAL study data from this paper are available to bona fide researchers from helpdesk@intervalstudy.org.uk and information, including the data access policy, is available at http://www.donorhealth-btru.nihr.ac.uk/project/bioresource.

## Code availability

Code using existing software is accessible via https://github.com/rxiangr/vQTL (Zenodo: https://sandbox.zenodo.org/account/settings/github/repository/rxiangr/vQTL; https://doi.org/10.5072/zenodo.187912). vQTL mapping used OSCA (v0.46): https://yanglab.westlake.edu.cn/software/osca/#Overview; genetic correlation analysis used LDSC (v1.01): https://github.com/bulik/ldsc; pleiotropy analysis: https://github.com/rondolab/HOPS. Mendelian randomisation used GSMR (v1.1.1): https://yanglab.westlake.edu.cn/software/gsmr/, MR-PRESSO: https://github.com/rondolab/MR-PRESSO and MendelianRandomisation (v0.1): https://cran.r-project.org/web/packages/MendelianRandomization/index.html; Analysis of selection used GCTB-BayesS (v2.05): https://cnsgenomics.com/software/gctb/#SummaryBayesianAlphabet; vPGS analysis used PRSICE-2 (v2.3.5): https://choishingwan.github.io/PRSice/ and plink2 (alpha4): https://www.cog-genomics.org/plink/2.0/; multi-trait GBLUP used SMTpred: https://github.com/uqrmaie1/smtpred; significance tests of $R^2$ increase used r2redux (v1.0.18): https://github.com/mommy003/r2redux; logistic regression analysis used glm(): https://www.rdocumentation.org/packages/stats/versions/3.6.2/topics/glm.

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

## Acknowledgements

We thank Dr. Emmanuela Bonglack and Dr. Xilin Jiang, for their insightful comments. The authors are grateful to Prof. Michael Goddard for discussions on the selection of vQTL. The authors are grateful to UK Biobank for access to data to undertake this study (Projects #30418). This work was supported by core funding from the: British Heart Foundation (RG/18/13/33946) and NIHR Cambridge Biomedical Research Centre (NIHR203312), Cambridge BHF Centre of Research Excellence (RE/18/1/34212) and BHF Chair Award (CH/12/2/29428) and by Health Data Research UK, which is funded by the UK Medical Research Council, Engineering and Physical Sciences Research Council, Economic and Social Research Council, Department of Health and Social Care (England), Chief Scientist Office of the Scottish Government Health and Social Care Directorates, Health and Social Care Research and Development Division (Welsh Government), Public Health Agency (Northern Ireland), British Heart Foundation and Wellcome. S.C.R. was funded by a British Heart Foundation (BHF) Programme Grant (RG/18/13/33946). S.C.R. was also funded by the National Institute for Health and Care Research (NIHR) Cambridge BRC (BRC-1215-20014; NIHR203312) [*]. Y.X. and M.I. were supported by the UK Economic and Social Research Council (ES/T013192/1). M.I. and S.L. were supported by NIH U24 (5U24HG012542-02). M.I. is supported by the Munz Chair of Cardiovascular Prediction and Prevention and the NIHR Cambridge Biomedical Research Centre (NIHR203312) [*]. *The views expressed are those of the authors and not necessarily those of the NIHR or the Department of Health and Social Care.

## Author contributions

R.X. and M.I. conceived of the study. R.X. performed the analyses with assistance from Y.X. and M.I.. R.X., M.I., F.T., Y.L., C.B.E., X.J., S.R., S.A.L., and Y.X. drafted and revised the manuscript. All authors read and approved the final version of the manuscript.

 

## Competing interests
