## [Transparent Peer Review file · Nature Communications]

Genome-wide analyses of variance in blood cell phenotypes provide new insights into complex trait biology and prediction

Corresponding Author: Professor Michael Inouye

Version 0:

Reviewer comments:

Reviewer #1

(Remarks to the Author)

The authors set out to conduct genome-wide vQTL analyses of almost thirty blood cell traits in the UK Biobank, looking to understand differences in the biological underpinnings and nature of selection on the variance as compared to the mean levels of these traits. They report a substantial number of vQTLs not previously identified in conventional GWAS analyses and provide evidence of negative selection against extreme trait variance. They then build variance-based polygenic scores (vPGS) and show that they augment standard PGS performance in the held-out INTERVAL cohort by defining strata across which there is differential PGS performance (also demonstrated via interaction testing). The authors also report associations between genetic predictors of blood cell trait variability and alcohol intake (both the variable itself and its genetic predictors), suggesting an effect of alcohol intake on this variability.

This study builds upon a growing literature on vQTLs, including but not limited to UKB studies, by moving into a new set of continuous trait outcomes. It expands in multiple valuable directions, including the consideration of genetic selection, the use of Mendelian randomization with variability-related outcomes, and an evaluation of the impact of vPGS on standard PGS performance (as a modifier for stratification/interaction). I have a few substantial comments about the vPGS construction and alcohol MR interpretation, as well as a series of more minor comments below. Regardless, I appreciate these new directions and think this is a solid contribution to the vQTL-specific and general genetic epidemiology literature that I expect to revisit and reference myself.

Major points:

- I appreciate the extensive exploration of vPGS performance and interaction, and think it's a great strength of this paper. With that said, I have some comments and concerns on the implementation and analysis:
 - o I'm unsure whether the "back-transformed" betas provided by OSCA provide a firm enough foundation for building PGS, given that the betas are derived from the significance of a non-additive statistical test combined with an effect direction that functionally does assume additivity. This could be solved by re-doing all vQTL analyses using a statistical approach for which "true" betas are available (e.g., DRM or QUAIL), but short of that, I could also imagine: (1) running one of these programs for one biomarker and demonstrating very strong similarity between the vPGS from OSCA vs. the alternate program, and/or (2) providing a theoretical justification for the validity of this choice.
 - o Is there a main effect of vPGS (not just interaction)? What would that mean?
 - o Minor: In the Discussion, there is wording indicating that vPGS-based stratification produces differential PGS performance while there is also a vPGSxPGS interaction. I might change the wording slightly to indicate that these are two ways of illustrating the same phenomenon (decreasing PGS-Y correlation as vPGS increases), rather than two separate observations.
 - o Because of the potential for quantile-specific heritability (ex., Williams 2022 <https://doi.org/10.1371/journal.pone.0262395> and many other studies by the same author), it would be great to see whether any of the PGS performance differences when stratifying by vPGS also show up when stratifying by the PGS itself.
 - o The authors base their vPGS creation strategy on that of Miao et al. 2022, who use PRSice-2 without p-value filtering. Is there any theoretical or empirical support for this choice? PRSice-2 is able to calculate PGS from multiple p-values thresholds that could be empirically compared for their association with phenotypic variability in a held-out dataset.
- I support the authors' decision to evaluate potential links between alcohol consumption and blood cell trait variance based

on the enrichment of vQTL near ALDH2, but have some concerns about the genetic instruments for alcohol intake. Since many variants associated with alcohol intake are mapped to genes involved in ethanol metabolism, it seems likely that much of their effect on behavior is downstream of their effects on physiology (see Woolf B et al., Am J Epi 2024: kwae143 for a more in-depth discussion of what they term “misidentifying the direction of the effect” with respect to genetic instruments for caffeine intake). With this in mind, can the authors provide any further results and/or commentary to support the causality of alcohol intake behavior?

Specific comments (in order of the manuscript):

- Intro (line 86): Consider rewording that sentence (“We found novel...”) for grammar/clarity.
- Results (line 128): Could be helpful to reinforce in this sentence that these two traits have negative genetic correlations between their levels and variances, rather than with each other. I know that is mentioned above, but this took me a few re-reads of the sentence to understand.
- Results – selection: I like this comparison of selection coefficients for variance vs. level when collapsing into a mean S_{level} and mean $S_{variance}$. When looking across traits, how correlated are selection coefficients for levels vs. variances? Put another way, are the traits whose levels are under the greatest negative selection also those whose variances are under the greatest negative selection?
- Results (line 152): typo (“MR-PRESSOR”)
- Discussion (and generally): There are a few different mentions in the manuscript of having discovered GxEs (e.g., line 263-5: “In conclusion, our study...and identification of GxE.”). Though the relevance of vQTLs to GxE discovery is clear, what specific GxE identification are the authors referring to?
- The Methods describe the identification of lead vQTLs (PLINK clumping) and their overlap with previously reported significant vQTLs (from Vuckovic). How did the authors define independence/overlap between lead vQTLs and (1) lead QTLs for levels of the same trait, and (2) lead vQTLs for other (e.g., for counting traits per locus reporting of pleiotropy)?
- Methods (line 293): There could be more detail describing the rationale for and coding of the “diet, smoking, and alcohol consumption” adjustment variables. If preprocessing followed precisely that of a specific previous paper (of the three that are cited; possibly Xu et al.?), then that could be stated explicitly, and it still might be helpful to give a bit more detail in this paper.
- Methods: How non-normally distributed are most of these blood biomarker traits? Was there any thought of log-transforming? This is a tricky issue and there is certainly simulation-based support in prior studies for avoiding log-transformation, but regardless it would be helpful to explicitly comment on this point given how common such transforms are in these types of studies.
- Methods: For the analysis of “genetically variable individuals”, what was the rationale behind binarizing at such extremes of vPGS (bottom/top 5%), which tosses out most of the sample? In this same analysis, what was the rationale for using never/former/current alcohol intake, rather than more standard categories (such as low/moderate/high drinks per week) or a continuous metric of consumption?
- It would be helpful to be explicit earlier in the manuscript (abstract and/or early results) about the use of a European ancestry-only sample, since the Methods come at the end.
- Figure 6: Can the units be standardized (say, OR per std. dev.) so the effect sizes are easier to visualize in the same plot?
- If possible, it would be great to make the analysis code accessible.

-Kenny Westerman

Reviewer #2

(Remarks to the Author)

Strengths:

1. Novel Approach: The study introduces and validates the concept of variance quantitative trait loci (vQTL) and variance polygenic scores (vPGS), which is an innovative approach in genetic studies. This adds a new dimension to understanding complex traits beyond traditional mean-focused analyses.
2. Comprehensive Analysis: The authors used a large dataset from the UK Biobank and the INTERVAL cohort, which lends significant statistical power and generalizability to their findings. The robust methodology, including Levene's test for vQTL mapping and Mendelian randomization, strengthens the validity of their results.

3. Practical Implications: The study's findings have potential clinical applications, particularly in improving genetic prediction accuracy. The idea of stratifying individuals based on vPGS to enhance polygenic risk score (PGS) predictions is a valuable contribution to precision medicine.

Criticisms and Areas for Improvement:

1. Complexity and Accessibility:

o Issue: The paper is highly technical and might be challenging for readers who are not specialists in quantitative genetics or statistical genomics. The complex statistical methods, such as Levene's test and Mendelian randomization, are discussed briefly without sufficient context for a broader audience.

o Recommendation: The authors could improve the accessibility of the paper by providing a clearer explanation of the statistical methods used, including their assumptions and limitations. Explaining these methods in layman's terms could be beneficial -- it's currently buried in the supplement but maybe discuss more in main text.

2. Interpretation of Results:

o Issue: While the identification of vQTLs and their stronger negative selection is intriguing, the biological interpretation of these findings is somewhat limited. The discussion on how these findings translate to biological or clinical relevance is not

as developed as it could be.

o Recommendation: The authors should elaborate on the biological implications of their findings. For example, they could explore potential mechanisms by which stronger selection on variance might occur and how this could impact disease phenotypes or treatment outcomes.

3. Validation and Replication:

o Issue: Although the study uses large datasets, the validation of vQTLs and vPGSs in additional, independent cohorts would strengthen the conclusions. The reliance on UK Biobank and INTERVAL cohorts, while robust, may not capture all genetic or environmental diversity.

o Recommendation: The authors could enhance their study by including replication in additional cohorts with different ethnic backgrounds or environmental exposures. This would test the generalizability of their findings and potentially uncover additional insights.

4. Gene-Environment Interaction Analysis:

o Issue: The study suggests gene-environment interactions (GxE) contribute to trait variance, but this is not explored in depth. The identification of alcohol consumption as a significant factor is interesting, but other potential environmental or lifestyle factors are not as thoroughly examined.

o Recommendation: A more comprehensive analysis of GxE interactions, including a broader range of environmental variables, would provide a more holistic understanding of factors influencing trait variance. Additionally, a deeper exploration of the interaction between vQTLs and specific environmental factors could uncover important insights.

5. Clinical Utility:

o Issue: The potential clinical utility of vPGS is highlighted, but practical considerations for implementation in clinical settings are not discussed. For example, the feasibility of integrating vPGS into existing genetic testing frameworks is not addressed.

o Recommendation: The authors could discuss the practical challenges and steps needed to translate vPGS findings into clinical practice. This might include the development of tools or guidelines for clinicians, as well as considerations for ethical and equitable implementation.

6. Ethical Considerations:

o Issue: The paper does not address the ethical implications of using vPGS in clinical practice, especially in terms of stratifying patients based on genetic variability.

o Recommendation: Including a discussion on the ethical implications of implementing vPGS in healthcare, particularly concerning patient consent, data privacy, and potential disparities in access, would be important. Addressing these issues could make the study more comprehensive and forward-thinking.

Recommendation:

Overall, this is a strong and innovative study that makes a significant contribution to the field of genetic epidemiology. The introduction of vQTLs and vPGSs offers a novel perspective on trait variance and genetic prediction. However, before publication, the paper would benefit from addressing the aforementioned criticisms, particularly in terms of expanding the discussion on biological implications, validating findings in diverse populations, and considering the practical and ethical aspects of clinical implementation.

If the authors can address these issues, I would recommend the paper for publication with minor revisions. If these areas remain unaddressed, more substantial revisions may be necessary to ensure the study's findings are both robust and broadly applicable.

Strengths:

1. Novel Approach: The study introduces and validates the concept of variance quantitative trait loci (vQTL) and variance polygenic scores (vPGS), which is an innovative approach in genetic studies. This adds a new dimension to understanding complex traits beyond traditional mean-focused analyses.

2. Comprehensive Analysis: The authors used a large dataset from the UK Biobank and the INTERVAL cohort, which lends significant statistical power and generalizability to their findings. The robust methodology, including Levene's test for vQTL mapping and Mendelian randomization, strengthens the validity of their results.

3. Practical Implications: The study's findings have potential clinical applications, particularly in improving genetic prediction accuracy. The idea of stratifying individuals based on vPGS to enhance polygenic risk score (PGS) predictions is a valuable contribution to precision medicine.

Criticisms and Areas for Improvement:

1. Complexity and Accessibility:

o Issue: The paper is highly technical and might be challenging for readers who are not specialists in quantitative genetics or statistical genomics. The complex statistical methods, such as Levene's test and Mendelian randomization, are discussed briefly without sufficient context for a broader audience.

o Recommendation: The authors could improve the accessibility of the paper by providing a clearer explanation of the statistical methods used, including their assumptions and limitations. Supplementary material or a brief appendix explaining these methods in layman's terms could be beneficial.

2. Interpretation of Results:

o Issue: While the identification of vQTLs and their stronger negative selection is intriguing, the biological interpretation of these findings is somewhat limited. The discussion on how these findings translate to biological or clinical relevance is not as developed as it could be.

o Recommendation: The authors should elaborate on the biological implications of their findings. For example, they could explore potential mechanisms by which stronger selection on variance might occur and how this could impact disease phenotypes or treatment outcomes.

3. Validation and Replication:

o Issue: Although the study uses large datasets, the validation of vQTLs and vPGSs in additional, independent cohorts would strengthen the conclusions. The reliance on UK Biobank and INTERVAL cohorts, while robust, may not capture all genetic or environmental diversity.

o Recommendation: The authors could enhance their study by including replication in additional cohorts with different ethnic

backgrounds or environmental exposures. This would test the generalizability of their findings and potentially uncover additional insights.

4. Gene-Environment Interaction Analysis:

o Issue: The study suggests gene-environment interactions (GxE) contribute to trait variance, but this is not explored in depth. The identification of alcohol consumption as a significant factor is interesting, but other potential environmental or lifestyle factors are not as thoroughly examined.

o Recommendation: A more comprehensive analysis of GxE interactions, including a broader range of environmental variables, would provide a more holistic understanding of factors influencing trait variance. Additionally, a deeper exploration of the interaction between vQTLs and specific environmental factors could uncover important insights.

5. Clinical Utility:

o Issue: The potential clinical utility of vPGS is highlighted, but practical considerations for implementation in clinical settings are not discussed. For example, the feasibility of integrating vPGS into existing genetic testing frameworks is not addressed.

o Recommendation: The authors could discuss the practical challenges and steps needed to translate vPGS findings into clinical practice. This might include the development of tools or guidelines for clinicians, as well as considerations for ethical and equitable implementation.

Reviewer #3

(Remarks to the Author)

Xiang et al. performed a genome-wide vQTL study for blood cell phenotypes -- they sought to identify variants that increase the likelihood that phenotype deviates from the mean (possibly because such genetic variants increase likelihood for individual to be affected by environments, and therefore higher phenotypic variance, or such variants affect other variables which directly change phenotype variance).

There are several conclusions from this work: vQTLs are largely independent from traditional additive QTLs; vQTLs are more susceptible by negative selections than additive QTLs; vQTLs can be used to identify individuals with more stable phenotypes and therefore increased PGS accuracy; alcohol consumption is correlated with genetic component of blood cell variances.

The conclusions are interesting and the chosen methods are appropriate -- I have a few questions that authors may find useful.

Comments:

1. A conclusion being made contrasting the selection coefficient between vQTLs and additive QTLs is "our results are consistent with evolution acting on blood cell traits to remove extreme phenotypes from the population." It can be argued both vQTLs and additive QTLs can push the phenotypes to the extreme, as vQTLs act to increase the phenotype variance, while additive QTLs act to increase the phenotype mean. I don't know if the quoted conclusion is made from the comparison of selection coefficient between vQTL and additive QTL, or just from the selection coefficient of vQTLs being significantly negative. More explanation is helpful.
2. The authors have sought to annotate and interpret vQTLs (e.g., Line 144-162). I suggest two additional analyses: (1) test whether vQTLs can be explained by GxE -- GxE terms can be included to see if vQTL effect can be explained away (2) instead of using genetic component of alcohol consumption, they can test whether the measured alcohol consumption increase the variance of blood cell phenotypes.
3. The authors reported increased variance explained by adding vPGS. This is interesting and surprising, since vPGS should not be able to predict phenotype mean. Can authors provide more explanation on this?
4. "Lifestyle effects on blood cell trait variance": instead correlating vPGS with alcohol consumption, can authors directly test whether observed variance of blood cell phenotypes is increased by alcohol consumption? They can also compare the strength of the correlation derived from either vPGS or observed variance.
5. A hypothesis lurking in the analysis related to alcohol consumption is that the identified vQTLs increase alcohol consumption, which in turn increases the variance of blood cell phenotypes. The authors have indeed performed mendelian randomization analysis to test this hypothesis. I think adding a genetic correlation analysis between alcohol consumption and vQTLs is useful as it uses genome-wide information (although can be less interpretable) -- a separate paragraph properly stating and testing this hypothesis is helpful if the authors agree.

Reviewer #4

(Remarks to the Author)

Version 1:

Reviewer comments:

Reviewer #1

(Remarks to the Author)

I appreciate the authors' extensive responses and updates to the manuscript in response to these reviews. To follow up on a few points:

- It's good to see that the correlation between this OSCA (Levene's) and DRM-based vPGS for MCV is fairly high. This result feels quite important, since the balance of the manuscript is based on vPGS calculated from these back-transformed betas, and I appreciate the authors including it.
- Quantile-specific heritability: It seems that my question may not have come across effectively. The authors say that "... because PGS is computed based on trait level, individuals ranked at the top in PGS would have higher prediction accuracy of PGS by definition." What I meant to suggest is that PGS prediction might be different in the subset of individuals with PGS percentile >90th, compared to the subset of individuals with PGS percentile <10th, for example. It isn't trivially true that PGS performance must be better in a subset of individuals with higher values for that same PGS (i.e., it is not true by definition). I still recommend checking results after this stratification (PGS performance within bins for that same PGS) as a direct comparison to the main results (PGS performance within bins for the vPGS). I agree with the authors that the "doubly stratified" result they describe in this response to reviewers, stratifying by vPGS within larger strata defined by PGS, does not necessarily add clear value to the existing manuscript.
- Alcohol and MR: Having reviewed the papers published by the larger group from Woolf et al., it is not clear what the authors are referencing in saying that that group has used ALDH2 variants in conducting MR analyses with alcohol as the exposure. I do appreciate the results now presented in Supp. Fig. 11 as additional support. However, the claims ("our results support the hypothesis that alcohol consumption increases variation in blood cell traits.") may warrant a bit more explicit discussion and literature support for the appropriateness of GSMR with alcohol intake as an exposure.
- The authors provide the following sentence for clarity about the procedure for determining locus novelty/independence: "As there are between trait correlations, a novel vQTL was defined as those lead vQTL after clumping with GWAS that did not have a p-value < 4.6×10^{-9} for any blood cell trait levels in Vuckovic et al and was not in strong LD ($r^2 < 0.8$) with a reported lead QTL for any trait in Vuckovic et al." What does "clumping with GWAS" mean here? In general, it might be valuable to elaborate on each step of the procedure that was undertaken to define novelty and locus overlap/counting.

Reviewer #2

(Remarks to the Author)

This manuscript provides significant contributions to statistical genetics by identifying 176 vQTLs across 29 blood cell traits, with 147 of these not overlapping with conventional additive QTLs. This underscores the importance of variance-based mapping in uncovering previously unexplored genetic loci.

The finding that vPGS can stratify individuals based on genetic variability and enhance the predictive accuracy of PGS is a novel and valuable addition to the field. Demonstrating that combining PGS with vPGS improves trait prediction by ~10% on average is particularly impactful for applications in personalized medicine.

The evidence of stronger negative selection on blood cell trait variance relative to trait levels offers an important evolutionary perspective, providing a nuanced understanding of the selective pressures shaping these phenotypes.

By integrating variance mapping with polygenic scores, this study extends the current understanding of genotype-to-phenotype relationships. The novel use of vPGS addresses an underexplored dimension of phenotypic prediction, advancing both theoretical and applied genetics.

These findings complement and expand prior research on vQTLs in traits like BMI and cardiometabolic biomarkers, but the focus on blood cell traits—ubiquitous in clinical testing—elevates their translational relevance.

I found the following real strengths in reading this manuscript:

The application of Levene's test to identify vQTLs is appropriate for variance mapping, and the use of LDSC to validate inflation factors and assess genetic correlations demonstrates high methodological standards.

Mendelian randomization (MR) analyses to evaluate causal effects of alcohol consumption on trait variance are well-conceived, although the biological plausibility and potential biases (e.g., pleiotropy) warrant additional discussion.

Stratifying individuals using vPGS and testing its impact on PGS predictive accuracy is a compelling demonstration of vPGS utility. However, reliance on PRSice for vPGS construction may limit scalability to non-European populations.

Data Analysis and Interpretation

The results support most of the conclusions, particularly the independence of vQTLs from additive QTLs and the predictive utility of vPGS. However, the interpretation of interactions between PGS and vPGS could be elaborated to clarify their implications for prediction accuracy and biological understanding.

The functional annotations of vQTLs, especially their enrichment in regulatory regions and association with environmental factors like alcohol consumption, are well-documented but could benefit from additional mechanistic insights.

The extensive methods section and availability of summary statistics and code are commendable, making the study highly reproducible. Clear documentation of assumptions (e.g., normality for Levene's test) further strengthens confidence in the analyses.

Following suggestions:

1) While the study highlights the utility of vPGS in stratifying individuals and improving prediction, additional discussion on its potential clinical applications (e.g., identifying high-risk subgroups, informing therapeutic strategies) would strengthen its translational impact.

Clarify Implications of Interactions:

The observed interactions between PGS and vPGS merit further discussion. Are these interactions primarily of methodological interest, or do they reveal biological phenomena that could refine risk prediction models?

) I think you could do a better job of discussing Limitations in Causal Inference.

The MR findings linking alcohol consumption to trait variance are intriguing but should be interpreted cautiously. Pleiotropy and potential confounders in MR analyses could be further addressed to contextualize these results.

3) The study's focus on individuals of European ancestry is a recognized limitation. Consider discussing how the methodology could be extended or adapted to diverse populations, given the pressing need for equity in polygenic prediction.

4) The study identifies stronger negative selection on variance than on mean levels, which is novel. Expanding on how these findings align with or differ from established theories of stabilizing selection could enrich the evolutionary narrative.

Reviewer #3

(Remarks to the Author)

The authors have sufficiently addressed all my previous comments.

Reviewer #4

(Remarks to the Author)

Version 2:

Reviewer comments:

Reviewer #1

(Remarks to the Author)

I appreciate the authors' continued attention to these comments and am in support of publication. I think the relationship between vPGS and PGS prediction accuracy is worthy of focused attention in future follow-up papers, and would leave it to the editor(s) to weigh in on the addition of these new figures/data (currently in the response to reviewers) to the current manuscript.

Reviewer #2

(Remarks to the Author)

After reviewing the authors' rebuttal to reviewer comments, I find they have satisfactorily addressed the major concerns raised during the review process. The authors have clarified their methodology for vQTL identification and novel vQTL definition, conducted the requested stratification analysis comparing PGS accuracy across different percentile bins, and expanded their discussion in several key areas including limitations of their Mendelian randomization analysis, potential clinical applications of vPGS, and the evolutionary context for their findings about negative selection on blood cell trait variance. While some responses could be more substantive—particularly regarding clinical applications and the biological interpretation of PGS-vPGS interactions—the overall quality of the revisions demonstrates a genuine effort to improve the manuscript based on expert feedback. The additional caution in interpreting causal relationships and the expanded methodological clarity have strengthened the manuscript considerably, making it a valuable contribution to our understanding of genetic architecture and phenotypic variation in blood cell traits.

Reviewer #4

(Remarks to the Author)

Point-to-point responses to reviewers re: NCOMMS-24-28845 “Genome-wide analyses of variance in blood cell phenotypes provide new insights into complex trait biology and prediction”

Thank you very much for comments/suggestions on our manuscript. Please see our response to your comments in blue text. Of note, line numbers in this document refer to the clean version of the manuscript.

REVIEWER COMMENTS

Reviewer #1 (Remarks to the Author):

The authors set out to conduct genome-wide vQTL analyses of almost thirty blood cell traits in the UK Biobank, looking to understand differences in the biological underpinnings and nature of selection on the variance as compared to the mean levels of these traits. They report a substantial number of vQTLs not previously identified in conventional GWAS analyses and provide evidence of negative selection against extreme trait variance. They then build variance-based polygenic scores (vPGS) and show that they augment standard PGS performance in the held-out INTERVAL cohort by defining strata across which there is differential PGS performance (also demonstrated via interaction testing). The authors also report associations between genetic predictors of blood cell trait variability and alcohol intake (both the variable itself and its genetic predictors), suggesting an effect of alcohol intake on this variability.

This study builds upon a growing literature on vQTLs, including but not limited to UKB studies, by moving into a new set of continuous trait outcomes. It expands in multiple valuable directions, including the consideration of genetic selection, the use of Mendelian randomization with variability-related outcomes, and an evaluation of the impact of vPGS on standard PGS performance (as a modifier for stratification/interaction). I have a few substantial comments about the vPGS construction and alcohol MR interpretation, as well as a series of more minor comments below. Regardless, I appreciate these new directions and think this is a solid contribution to the vQTL-specific and general genetic epidemiology literature that I expect to revisit and reference myself.

Author response: Thank you for your positive comments on the manuscript. Please see our responses in detail in the following paragraphs.

Major points:

- I appreciate the extensive exploration of vPGS performance and interaction, and think it's a great strength of this paper. With that said, I have some comments and concerns on the implementation and analysis:
- I'm unsure whether the "back-transformed" betas provided by OSCA provide a firm enough foundation for building PGS, given that the betas are derived from the significance of a non-additive statistical test combined with an effect direction that functionally does assume additivity. This could be solved by re-doing all vQTL analyses using a statistical approach for which "true" betas are available (e.g., DRM or QUAIL), but short of that, I could also imagine: (1) running one of these programs for one biomarker and demonstrating very strong similarity between the vPGS from OSCA vs. the alternate program, and/or (2) providing a theoretical justification for the validity of this choice.

Author response: As requested, we have reanalysed the data of a biomarker (mcv, mean corpuscular volume), which was key to our alcohol consumption findings, using DRM and compared its vPGS with the vPGS computed using OSCA. The correlation of the vPGSs was very strong (0.904; Supplementary Figure 1). While we appreciate and would strongly support a separate methods-focused paper benchmarking various vQTL/vPGS methods, we believe that this would be outside the scope of the current study, which focuses on proof-of-concept for vQTL, vPGS and their GxE in blood cell traits. To articulate this, we have added our findings to the manuscript (Results line 97-100; Supplementary Figure 1).

- Is there a main effect of vPGS (not just interaction)? What would that mean?

Author response: Thank you for this comment. To address it, we have included additional results to the new Supplementary Table 11 to describe the full results of the interaction between PGS and vPGS, and we have added the following text to the Results (lines 223-226): "For seven traits (eo, rbc, plt, neut, mcv, baso and lymph, Supplementary Table 11), the main effects of their corresponding vPGS were also significant. As expected, the effects of vPGS were much smaller than PGS on trait levels, as the PGSs are directly estimated from trait levels."

While we hesitate to go too far beyond the observation and speculate as to its meaning, we believe that these findings are purely statistical, rather than biological, indicating for these seven traits there is some small amount of residual trait level variance that can be captured by vPGS (similar to ensembling gains from multiple PGS methods).

Minor:

- In the Discussion, there is wording indicating that vPGS-based stratification produces differential PGS performance while there is also a vPGSxPGS interaction. I might change the wording slightly to indicate that these are two ways of illustrating the same phenomenon (decreasing PGS-Y correlation as vPGS increases), rather than two separate observations.

Author response: We agree with these comments and have restructured the paragraph (lines 294-297) to indicate that some groups are intrinsically harder to predict by PGSs than others: Illustrating this point, our analysis found multiple significant interactions between PGS and vPGS, suggesting that the non-additive and GxE components related to PGSs could impact prediction accuracy. These findings are consistent with previous observations^{50,51} and may be important for PGS translation.

- Because of the potential for quantile-specific heritability (ex., Williams 2022 <https://doi.org/10.1371/journal.pone.0262395> and many other studies by the same author), it would be great to see whether any of the PGS performance differences when stratifying by vPGS also show up when stratifying by the PGS itself.

Author response: We thank you for the suggestion; however, we are confused by inconsistencies underpinning this idea. First, because PGS is computed based on trait level, individuals ranked at the top in PGS would have higher prediction accuracy of PGS by definition. Second, stratifying individuals by both PGS and vPGS groups reduces the sample size in each stratum, causing reduced and variable power.

To illustrate this, we performed the following analysis: We first stratified individuals into 3 PGS groups (high, medium and low), then stratified each PGS group into 10 vPGS strata (this way approximately 2000+ INTERVAL individuals are included in each stratum). We then compared the PGS accuracy between the bottom and top 20% of vPGS groups within each PGS strata. Consistent with our original results, the mean PGS accuracy is always higher in the bottom vPGS group compared to the top vPGS group, although such difference is only statistically significant within the high PGS individuals (see figure below). It's not clear what these results add to our existing analysis, thus we present them here in the response letter though, depending on editorial guidance, we are open to including them in the supplementary materials.

Figure legend: Comparison of PGS prediction accuracy between vPGS and PGS strata. High, medium and low PGS groups are individuals ranked the top, middle and bottom 1/3 of their PGS of blood cell traits, respectively. Bottom and top vPGS groups are individuals ranked top and bottom 20% within each PGS strata. **: $P < 0.01$. ns: not significant.

- The authors base their vPGS creation strategy on that of Miao et al. 2022, who use PRSice-2 without p-value filtering. Is there any theoretical or empirical support for this choice? PRSice-2 is able to calculate PGS from multiple p-values thresholds that could be empirically compared for their association with phenotypic variability in a held-out dataset.

Author response: We thank you for this comment. As mentioned above, the scope of this study is to establish proof-of-concept for vQTL, vPGS and GxE in blood cell traits and the benchmarking/optimisation of vPGS methods is a worthy follow-up study. In demonstrating the validity of vPGS, we used a simple well-established method, PRSice-2, in calculating PGSs that has been used by Miao et al, 2022. We note that using Pruning+Thresholding to develop the PGS could lead to underpowered PGS (Ni, Zeng et al. 2021, Wang, Namba et al. 2023) which in fact gives a conservative estimate of their effects. To clarify this for the reader, we have modified our limitations paragraph (lines 304-307) to include: “Our vPGS was computed using parameters implemented in PRSice-2³⁶ and reported by Miao et al¹⁶, which may provide a conservative estimate of the effects of vPGSs as PGSs developed by pruning+thresholding are usually underpowered^{54,55}.”

- I support the authors’ decision to evaluate potential links between alcohol consumption and blood cell trait variance based on the enrichment of vQTL near ALDH2, but have some concerns about the genetic instruments for alcohol intake. Since many variants associated with alcohol intake are mapped to genes involved in

ethanol metabolism, it seems likely that much of their effect on behavior is downstream of their effects on physiology (see Woolf B et al., Am J Epi 2024: kwae143 for a more in-depth discussion of what they term “misidentifying the direction of the effect” with respect to genetic instruments for caffeine intake). With this in mind, can the authors provide any further results and/or commentary to support the causality of alcohol intake behavior?

Author response: We note that the authors of the Woolf et al paper have themselves performed MR of alcohol consumption using *ALDH2* and similar genetic instruments which we utilise here [e.g., (Larsson, Burgess et al. 2020, Kassaw, Zhou et al. 2024)]. Furthermore, it is well known that both acetaldehyde and ethanol are toxic by-products of alcohol metabolism; therefore, it is not clear to us why one would favour genes involved in the production of acetaldehyde but not ethanol. The senior author of Woolf et al has noted as much previously, e.g., (Kassaw, Zhou et al. 2024). Given the consistency of our approach with that of Woolf et al coauthors as well as other previously published MR studies for alcohol consumption (Topiwala, Taschler et al. 2022, Kassaw, Zhou et al. 2024, Zheng, Liao et al. 2024), we are confident that our approach is sufficient to claim MR support for the causal effects of alcohol consumption. As with all computational methods for estimating causal effects, it is important to note that lab/clinical validation is ultimately an important part of triangulating causal evidence; thus, we have added this to the limitation paragraph before the conclusion (lines 313-315).

Specific comments (in order of the manuscript):

- Intro (line 86): Consider rewording that sentence (“We found novel...”) for grammar/clarity.

Author response: We have revised the text accordingly.

- Results (line 128): Could be helpful to reinforce in this sentence that these two traits have negative genetic correlations between their levels and variances, rather than with each other. I know that is mentioned above, but this took me a few re-reads of the sentence to understand.

Author response: We have revised this part of the text and the following sentence accordingly.

- Results – selection: I like this comparison of selection coefficients for variance vs. level when collapsing into a mean S_{level} and mean S_{variance} . When looking across traits, how correlated are selection coefficients for levels vs. variances? Put another way, are the traits whose levels are under the greatest negative selection also those whose variances are under the greatest negative selection?

Author response: Thank you for this suggestion. We have now performed the analysis suggested and the results are shown in Supplementary Figure 5. The Pearson correlation between S_level and mean S_variance was 0.143 ($p=0.46$) and the Spearman ranking correlation was 0.139 ($p=0.47$). Therefore, their correlation is positive but not significant.

- Results (line 152): typo (“MR-PRESSOR”)

Author response: Apologies for the typo, this has been corrected.

- Discussion (and generally): There are a few different mentions in the manuscript of having discovered GxEs (e.g., line 263-5: “In conclusion, our study...and identification of GxE.”). Though the relevance of vQTLs to GxE discovery is clear, what specific GxE identification are the authors referring to?

Author response: We have revised this section to clarify that “..., such as the effect of alcohol consumption is genetically linked to blood cell variances.” (lines 318-319). We have also conducted additional analyses for targeted GxE analysis including testing the effect of interactions between lead vQTLs and environmental factors of age, alcohol consumption, BMI, sex and smoking; these results are now included in the revised manuscript (lines 158-161, Supplementary Figure 6).

- The Methods describe the identification of lead vQTLs (PLINK clumping) and their overlap with previously reported significant vQTLs (from Vuckovic). How did the authors define independence/overlap between lead vQTLs and (1) lead QTLs for levels of the same trait, and (2) lead vQTLs for other (e.g., for counting traits per locus reporting of pleiotropy)?

Author response: As mentioned, there are trait-wise correlations and SNP-wise correlations so we keep the definition more conservative (not passing the significance threshold and not in strong LD with any lead QTL reported by trait in Vuckovic et al) as described in the revised manuscript which is also described here: “As there are between trait correlations, a novel vQTL was defined as those lead vQTL after clumping with GWAS that did not have a p -value $< 4.6 \times 10^{-9}$ for any blood cell trait levels in Vuckovic et al and was not in strong LD ($r^2 < 0.8$) with a reported lead QTL for any trait in Vuckovic et al.” (line 374-377).

- Methods (line 293): There could be more detail describing the rationale for and coding of the “diet, smoking, and alcohol consumption” adjustment variables. If preprocessing followed precisely that of a specific previous paper (of the three that

are cited; possibly Xu et al.), then that could be stated explicitly, and it still might be helpful to give a bit more detail in this paper.

Author response: We have added more description in this section: “The rationale for such adjustments was detailed in Astle et al ⁵. Briefly, after adjustment for age, sex, BMI and variables measuring smoking habits and alcohol consumption, covariates still explained $\geq 0.5\%$ of blood cell trait variance. Therefore, all relevant environmental variables were included in the adjustments.” (lines 347-351).

- **Methods:** How non-normally distributed are most of these blood biomarker traits? Was there any thought of log-transforming? This is a tricky issue and there is certainly simulation-based support in prior studies for avoiding log-transformation, but regardless it would be helpful to explicitly comment on this point given how common such transforms are in these types of studies.

Author response: We clarify that no log transformation was applied to the data due to preprocessing (from Astle et al) which minimised non-normality. To address your comments we conducted additional analyses and the following was added to Methods (lines 351-354) and Supplementary Table 12: “We used the Kolmogorov-Smirnov Test to check the normality of phenotypes where the null hypothesis is the data comes from a normal distribution. The smallest p-value was 0.82 (Supplementary Table 12) so all traits are expected to be normally distributed after quality control done previously.”

- **Methods:** For the analysis of “genetically variable individuals”, what was the rationale behind binarizing at such extremes of vPGS (bottom/top 5%), which tosses out most of the sample? In this same analysis, what was the rationale for using never/former/current alcohol intake, rather than more standard categories (such as low/moderate/high drinks per week) or a continuous metric of consumption?

Author response: The stratification of PGS into extreme bins is common in the field, and the conventional (though arbitrary) cut-offs are 1%, 5% and 10%. To be consistent with this convention, we considered these thresholdings for vPGS. The rationale for the selection of threshold was that INTERVAL (N~40,000) was not large enough to use a 1% threshold as this would lead to too few individuals in each group; thus, the threshold was loosened to 5%. For completeness, we have added the results for the top/bottom 10% (Supplementary Figure 12). As this result is highly similar to the 5% choice, we used results from 5% in the main results. The choice of never/former/current alcohol intake is due to the availability of the data. The other available data is alcohol intake which is also categorical (drinking most days / 1-2 times a week / 3-5 times a week / 1-3 times a month). Again for

completeness, we have analysed such data which achieved similar results (Supplementary Figure 13) from the analysis of never/former/current alcohol intake.

- It would be helpful to be explicit earlier in the manuscript (abstract and/or early results) about the use of a European ancestry-only sample, since the Methods come at the end.

Author response: To make this clearer for the reader, we have noted the European ancestry only in the last paragraph of the Introduction (lines 83).

- Figure 6: Can the units be standardized (say, OR per std. dev.) so the effect sizes are easier to visualize in the same plot?

Author response: We have revised **Figure 6** (displaying Z-score) accordingly, so effect sizes are on the same scale.

- If possible, it would be great to make the analysis code accessible.

Author response: Thank you for this suggestion. We have provided code access via <https://github.com/rxiangr/vQTL>.

Reviewer #2 (Remarks to the Author):

Strengths:

1. **Novel Approach:** The study introduces and validates the concept of variance quantitative trait loci (vQTL) and variance polygenic scores (vPGS), which is an innovative approach in genetic studies. This adds a new dimension to understanding complex traits beyond traditional mean-focused analyses.
2. **Comprehensive Analysis:** The authors used a large dataset from the UK Biobank and the INTERVAL cohort, which lends significant statistical power and generalizability to their findings. The robust methodology, including Levene's test for vQTL mapping and Mendelian randomization, strengthens the validity of their results.
3. **Practical Implications:** The study's findings have potential clinical applications, particularly in improving genetic prediction accuracy. The idea of stratifying individuals based on vPGS to enhance polygenic risk score (PGS) predictions is a valuable contribution to precision medicine.

Author response: Thank you for your positive comments on the manuscript. Please see our responses in detail in the following paragraphs.

Criticisms and Areas for Improvement:

1. Complexity and Accessibility:

o Issue: The paper is highly technical and might be challenging for readers who are not specialists in quantitative genetics or statistical genomics. The complex statistical methods, such as Levene's test and Mendelian randomization, are discussed briefly without sufficient context for a broader audience.

o Recommendation: The authors could improve the accessibility of the paper by providing a clearer explanation of the statistical methods used, including their assumptions and limitations. Explaining these methods in layman's terms could be beneficial -- it's currently buried in the supplement but maybe discuss more in main text.

Author response: We apologise if our communication veered too technical. In the revised manuscript, we have provided more context when introducing methods, including Levene's test, selection coefficient, FUMA, Mendelian randomisation and polygenic scores, for the first time in the results (e.g. lines 96, 141-142, 152-153, 166-168, and 199-200). We have also expanded on explanations in the Methods (e.g. lines 347-354, 374-377, 396-398). We hope that these clarifications improve the readability of our manuscript.

2. Interpretation of Results:

o Issue: While the identification of vQTLs and their stronger negative selection is intriguing, the biological interpretation of these findings is somewhat limited. The discussion on how these findings translate to biological or clinical relevance is not as developed as it could be.

o Recommendation: The authors should elaborate on the biological implications of their findings. For example, they could explore potential mechanisms by which stronger selection on variance might occur and how this could impact disease phenotypes or treatment outcomes.

Author response: While we would caution that the clinical relevance of our findings is indeed limited (rather there would be potential relevance if in future conventional polygenic scores themselves were available as part of routine healthcare), we have taken on board the reviewer's comments regarding biological implications and expanded our Discussion to include examples of potential mechanisms, particularly for neutrophils, explaining selection on blood cell trait variance (lines 270-282):

“Selection to reduce phenotypic variances implies stabilising selection for blood cell traits, which has been reported on other human traits¹⁴. For example, we observed stabilising selection for neutrophil percentage of white blood cells was strongest in people with genetically elevated neutrophil levels. Neutrophils are innate immune cells that act as first responders against infection by releasing cytotoxic antimicrobial peptides; damaging proximal tissue at their site of activity³⁹. Increased neutrophil abundance and activity are associated with myriad chronic inflammatory conditions and are predictive of long-term risk of cardiovascular risk and mortality^{40,41}. Importantly, hyperactivity of the innate immune response from severe infection events (such as COVID-19) can lead to “cytokine storms”; causing extensive tissue damage and rapidly leading to organ failure and death⁴². Therefore, selection to reduce extremely high levels of neutrophils may reflect evolutionary mechanisms acting on immune systems to improve survival.”

3. Validation and Replication:

o Issue: Although the study uses large datasets, the validation of vQTLs and vPGSs in additional, independent cohorts would strengthen the conclusions. The reliance on UK Biobank and INTERVAL cohorts, while robust, may not capture all genetic or environmental diversity.

o Recommendation: The authors could enhance their study by including replication in additional cohorts with different ethnic backgrounds or environmental exposures. This would test the generalizability of their findings and potentially uncover additional insights.

Author response: While we agree that it is important to assess the demographic and environmental generalisation of any research finding and have been part of major past efforts in human genomics, the amount and complexity of additional work beyond establishing the initial finding makes assessing demographic and environmental generalisability beyond the scope of this study. Our current analysis already includes external validations in the two large biobanks (INTERVAL and UK Biobank) which have comprised the vast majority of the individuals in the original blood cell trait GWASs (Astle et al, *Cell* 2016; Vuckovic et al, *Cell* 2020), thus we are confident our findings are robust. We anticipate that the additional years of research it would take to establish collaborations and to perform and interpret vQTL/vPGS analyses along these lines with multi-ethnic, multi-environmental exposure cohorts of sufficient size would constitute one, and potentially multiple, substantive subsequent studies.

4. Gene-Environment Interaction Analysis:

o Issue: The study suggests gene-environment interactions (GxE) contribute to trait variance, but this is not explored in depth. The identification of alcohol consumption

as a significant factor is interesting, but other potential environmental or lifestyle factors are not as thoroughly examined.

o Recommendation: A more comprehensive analysis of GxE interactions, including a broader range of environmental variables, would provide a more holistic understanding of factors influencing trait variance. Additionally, a deeper exploration of the interaction between vQTLs and specific environmental factors could uncover important insights.

Author response: We thank the reviewer for their comment. While we have been cautious in selecting environmental factors with well-established genetic instruments thus minimising the risk of commonly flawed MR analyses, e.g., (Haycock, Burgess et al. 2016, Burgess, Smith et al. 2019, Gkatzionis and Burgess 2019), we have expanded our GxE analyses to assess a broader range of exposures. Using identified lead vQTLs, we test the effects of interaction between them and five environmental factors: age, alcohol consumption, BMI, sex and smoking [Results (lines 158-161), Methods (lines 382-395) and Supplementary Figure 6]. We found many significant GxE signals from such analysis (study FDR < 5.5×10^{-5}) and also found alcohol consumption had one of the largest numbers of significant GxE associations, which provides additional support for the links between alcohol consumption identified by other analyses as described above.

5. Clinical Utility:

o Issue: The potential clinical utility of vPGS is highlighted, but practical considerations for implementation in clinical settings are not discussed. For example, the feasibility of integrating vPGS into existing genetic testing frameworks is not addressed.

o Recommendation: The authors could discuss the practical challenges and steps needed to translate vPGS findings into clinical practice. This might include the development of tools or guidelines for clinicians, as well as considerations for ethical and equitable implementation.

Author response: We thank you for considering the clinical potential of vPGS. However, we believe our current study only provides proof-of-concept evidence for vPGS to be informative on top of conventional PGSs. Therefore, we believe that any consideration of using vPGS in clinical practice is premature, pending the rigorous studies of implementation research including health economic evaluations in the clinical settings. These studies are of course out of scope for the current work but we wish to explore this in the future. However, we have added the following to the limitation paragraph (lines 309-313): “While our study provides proof-of-concept evidence for vPGS to be informative on top of conventional PGSs, any potential future role in clinical practice will depend on myriad factors, including the

infrastructure to deploy even conventional PGSs, quantification of clinical utility, and assessments of demographic transferability.”

6. Ethical Considerations:

- o Issue: The paper does not address the ethical implications of using vPGS in clinical practice, especially in terms of stratifying patients based on genetic variability.
- o Recommendation: Including a discussion on the ethical implications of implementing vPGS in healthcare, particularly concerning patient consent, data privacy, and potential disparities in access, would be important. Addressing these issues could make the study more comprehensive and forward-thinking.

Author response: Thank you for raising this point. As previously noted, we believe that any consideration of using vPGS in clinical practice is premature. While our study provides proof-of-concept, opening up the field to future investigations of blood cell trait variability and vPGS, there is much more research needed to establish clinical validity and clinical utility prior to investigating aspects of implementation science (and indeed health economic evaluations). While *prima facie*, we believe in principle there is little substantive difference in the ethics of potential use of vPGS as compared to conventional PGS, there is a need for further research to establish the relative clinical benefits vs harms before a responsible ethics discussion is possible. Without this, a discussion of the ethics of clinical implementation is premature.

Reviewer #3 (Remarks to the Author):

Xiang et al. performed a genome-wide vQTL study for blood cell phenotypes -- they sought to identify variants that increase the likelihood that phenotype deviates from the mean (possibly because such genetic variants increase likelihood for individual to be affected by environments, and therefore higher phenotypic variance, or such variants affect other variables which directly change phenotype variance).

There are several conclusions from this work: vQTLs are largely independent from traditional additive QTLs; vQTLs are more susceptible by negative selections than additive QTLs; vQTLs can be used to identify individuals with more stable phenotypes and therefore increased PGS accuracy; alcohol consumption is correlated with genetic component of blood cell variances.

The conclusions are interesting and the chosen methods are appropriate -- I have a few questions that authors may find useful.

Author response: Thank you for your supportive comments on the manuscript. Please see our responses detailed in the following paragraphs.

Comments:

1. A conclusion being made contrasting the selection coefficient between vQTLs and additive QTLs is "our results are consistent with evolution acting on blood cell traits to remove extreme phenotypes from the population." It can be argued both vQTLs and additive QTLs can push the phenotypes to the extreme, as vQTLs act to increase the phenotype variance, while additive QTLs act to increase the phenotype mean. I don't know if the quoted conclusion is made from the comparison of selection coefficient between vQTL and additive QTL, or just from the selection coefficient of vQTLs being significantly negative. More explanation is helpful.

Author response: Thank you for pointing this out. We agree that the clarity of this sentence should be improved and have revised this sentence as "While it can be difficult to differentiate between negative and stabilising selection, our results indicate negative selection is acting on both vQTLs and additive QTLs (somewhat more so on the former than the latter) to remove extreme blood cell phenotypes from the population." (lines 148-151).

2. The authors have sought to annotate and interpret vQTLs (e.g., Line 144-162). I suggest two additional analyses: (1) test whether vQTLs can be explained by GxE -- GxE terms can be included to see if vQTL effect can be explained away (2) instead of using genetic component of alcohol consumption, they can test whether the measured alcohol consumption increases the variance of blood cell phenotypes.

Author response: Thank you for this comment. As part of our response to Reviewer 2, comment 4, we have conducted targeted GxE analysis using lead vQTLs together with five environmental factors: age, alcohol consumption, BMI, sex and smoking [Results (lines 158-162), Methods (lines 382-395) and Supplementary Figure 6]. We found many significant GxE signals from this analysis (study FDR < 5.5×10^{-5}) and also found alcohol consumption had one of the largest numbers of significant GxE associations, consistent with our previous analyses linking alcohol consumption and blood cell trait variances.

We also linked the phenotypic variance in blood cell traits with alcohol consumption by doing a Levene's test in the UK Biobank data (Supplementary Figure 11). We saw alcohol consumption is associated with increased variances in many blood cell phenotypes, including red blood cell counts (rbc), mean spheric corpuscular volume (mscv), mean corpuscular volume (mcv) and neutrophil percentage of white cells

(neut_p) which were supported by Mendelian randomisation analyses. Therefore, the additional analysis supported our conclusions.

3. The authors reported increased variance explained by adding vPGS. This is interesting and surprising, since vPGS should not be able to predict phenotype mean. Can authors provide more explanation on this?

Author response: We have revised the Discussion (lines 292-297) to add more explanation: "Stratification by vPGS was shown to identify groups with significantly different PGS prediction accuracy, indicating that some groups are intrinsically harder to predict by PGSs than others. Illustrating this point, our analysis found multiple significant interactions between PGS and vPGS, suggesting that the non-additive and GxE components related to PGSs could impact prediction accuracy. These findings are consistent with previous observations^{50,51} and may be important for PGS translation."

4. "Lifestyle effects on blood cell trait variance": instead correlating vPGS with alcohol consumption, can authors directly test whether observed variance of blood cell phenotypes is increased by alcohol consumption? They can also compare the strength of the correlation derived from either vPGS or observed variance.

Author response: As requested, we have tested the association of self-reported alcohol consumption with blood cell phenotypes using Levene's test (Supplementary Figure 11). Consistent with our MR results for genetically predicted alcohol consumption, self-reported alcohol consumption was associated with multiple blood cell phenotypes including mean corpuscular volume (mcv) and neutrophil percentage of white cells (neut_p). We omit a direct comparison of the effects of vPGS and alcohol consumption, as the alcohol consumption measurement at baseline does not have a causal interpretation due to the blood cell phenotypes having also been measured at baseline.

5. A hypothesis lurking in the analysis related to alcohol consumption is that the identified vQTLs increase alcohol consumption, which in turn increases the variance of blood cell phenotypes. The authors have indeed performed mendelian randomization analysis to test this hypothesis. I think adding a genetic correlation analysis between alcohol consumption and vQTLs is useful as it uses genome-wide information (although can be less interpretable) -- a separate paragraph properly stating and testing this hypothesis is helpful if the authors agree.

Author response: Thank you for this comment. We have followed the suggestion to compute genetic correlations between alcohol consumption and variance of blood cell traits and compared these effects with the effects from Mendelian randomisation between alcohol consumption and variance of blood cell traits, which confirm our previous observations (lines 161-163, 179-181, Supplementary Figure 7, Supplementary Table 10).

References:

- Burgess, S., G. D. Smith, N. M. Davies, F. Dudbridge, D. Gill, M. M. Glymour, F. P. Hartwig, Z. Kutalik, M. V. Holmes and C. Minelli (2019). "Guidelines for performing Mendelian randomization investigations: update for summer 2023." Wellcome open research **4**.
- Gkatzionis, A. and S. Burgess (2019). "Contextualizing selection bias in Mendelian randomization: how bad is it likely to be?" International journal of epidemiology **48**(3): 691-701.
- Haycock, P. C., S. Burgess, K. H. Wade, J. Bowden, C. Relton and G. D. Smith (2016). "Best (but oft-forgotten) practices: the design, analysis, and interpretation of Mendelian randomization studies." The American journal of clinical nutrition **103**(4): 965-978.
- Kassaw, N. A., A. Zhou, A. Mulugeta, S. H. Lee, S. Burgess and E. Hyppönen (2024). "Alcohol consumption and the risk of all-cause and cause-specific mortality—a linear and nonlinear Mendelian randomization study." International journal of epidemiology **53**(2): dyae046.
- Larsson, S. C., S. Burgess, A. M. Mason and K. Michaëlsson (2020). "Alcohol consumption and cardiovascular disease: a Mendelian randomization study." Circulation: Genomic and Precision Medicine **13**(3): e002814.
- Ni, G., J. Zeng, J. A. Revez, Y. Wang, Z. Zheng, T. Ge, R. Restuadi, J. Kiewa, D. R. Nyholt and J. R. Coleman (2021). "A comparison of ten polygenic score methods for psychiatric disorders applied across multiple cohorts." Biological psychiatry **90**(9): 611-620.
- Topiwala, A., B. Taschler, K. Ebmeier, S. Smith, H. Zhou, D. Levey, V. Codd, N. Samani, J. Gelernter and T. Nichols (2022). "Alcohol consumption and telomere length: Mendelian randomization clarifies alcohol's effects." Molecular Psychiatry **27**(10): 4001-4008.
- Wang, Y., S. Namba, E. Lopera, S. Kerminen, K. Tsuo, K. Läll, M. Kanai, W. Zhou, K.-H. Wu and M.-J. Favé (2023). "Global Biobank analyses provide lessons for developing polygenic risk scores across diverse cohorts." Cell Genomics **3**(1).
- Zheng, L., W. Liao, S. Luo, B. Li, D. Liu, Q. Yun, Z. Zhao, J. Zhao, J. Rong and Z. Gong (2024). "Association between alcohol consumption and incidence of dementia in current drinkers: linear and non-linear mendelian randomization analysis." EClinicalMedicine **76**.

Point-to-point responses to reviewers (R2) re: NCOMMS-24-28845 “Genome-wide analyses of variance in blood cell phenotypes provide new insights into complex trait biology and prediction”

Reviewer #1 (Remarks to the Author):

I appreciate the authors' extensive responses and updates to the manuscript in response to these reviews. To follow up on a few points:

- It's good to see that the correlation between this OSCA (Levene's) and DRM-based vPGS for MCV is fairly high. This result feels quite important, since the balance of the manuscript is based on vPGS calculated from these back-transformed betas, and I appreciate the authors including it.

Thank you for this suggestion which has improved the manuscript.

- Quantile-specific heritability: It seems that my question may not have come across effectively. The authors say that "...because PGS is computed based on trait level, individuals ranked at the top in PGS would have higher prediction accuracy of PGS by definition." What I meant to suggest is that PGS prediction might be different in the subset of individuals with PGS percentile >90th, compared to the subset of individuals with PGS percentile <10th, for example. It isn't trivially true that PGS performance must be better in a subset of individuals with higher values for that same PGS (i.e., it is not true by definition). I still recommend checking results after this stratification (PGS performance within bins for that same PGS) as a direct comparison to the main results (PGS performance within bins for the vPGS). I agree with the authors that the "doubly stratified" result they describe in this response to reviewers, stratifying by vPGS within larger strata defined by PGS, does not necessarily add clear value to the existing manuscript.

We thank the reviewer for further clarification of their query. We have updated the analysis based on your comments (see below figure). Essentially, as expected, the results show the accuracy of PGSs are best for the top and bottom PGS groups. The top vPGS group has a somewhat higher PGS accuracy than the bottom vPGS group which has been shown in the main and supplementary figures. As this analysis does not substantially inform the conclusions of the manuscript, we present them here in the response letter but have not currently added them to the main/supplementary text; however, we would seek editorial guidance as to whether and, if so, how to include it given relevance and space considerations.

Figure: Comparing the accuracy of PGS of blood cell traits across PGS bins (a) and vPGS bins (b). Accuracy is the Pearson correlation between PGS and blood cell trait. The INTERVAL cohort was stratified by PGS bins (a) and vPGS bins (b) with percentile (0,10] being the highest PGS value (i.e. most variable vPGS) and percentile (90,100] being the lowest. In each bin, the box represents the distribution of the PGS prediction accuracy across 26 blood cell traits. Blue dashed lines represent the averaged PGS accuracy between 10 bins of PGS (a) and vPGS (b).

- Alcohol and MR: Having reviewed the papers published by the larger group from Woolf et al., it is not clear what the authors are referencing in saying that that group has used ALDH2 variants in conducting MR analyses with alcohol as the exposure. I do appreciate the results now presented in Supp. Fig. 11 as additional support. However, the claims (“our results support the hypothesis that alcohol consumption increases variation in blood cell traits.”) may warrant a bit more explicit discussion and literature support for the appropriateness of GSMR with alcohol intake as an exposure.

We thank you for reviewing this point with extra care and appreciating our efforts in revising the manuscript. There is both new and established evidence to show that alcohol drinking causes abnormal blood cell phenotypes. Apart from GSMR, we used additional Mendelian randomisation methods (weighted-median and MR-PRESSO, lines 168-169, Figure 2b and Supplementary Table 10) to validate our work. We have also added the following text to the Discussion (lines 311-319):

“In the Mendelian Randomisation analysis, we have chosen alcohol consumption as the exposure as established evidence supports the adverse effects of alcohol drinking on blood

cell morphologies^{56,58,59}, likely due to mediation by inflammation and immune responses⁶⁰. To ensure correction of potential pleiotropic confounders we used GSMR as a discovery tool for our Mendelian Randomisation analyses and verification with MR-PRESSO and weighted median. While GSMR and MR-PRESSO both correct for pleiotropic confounders, there could exist other confounders not accounted for in the current study, thus we caution that this is evidence for, but not proof of, a causal interpretation of the effects of alcohol consumption on blood cell trait variances.”

- The authors provide the following sentence for clarity about the procedure for determining locus novelty/independence: “As there are between trait correlations, a novel vQTL was defined as those lead vQTL after clumping with GWAS that did not have a p-value $< 4.6 \times 10^{-9}$ for any blood cell trait levels in Vuckovic et al and was not in strong LD ($r^2 < 0.8$) with a reported lead QTL for any trait in Vuckovic et al.” What does “clumping with GWAS” mean here? In general, it might be valuable to elaborate on each step of the procedure that was undertaken to define novelty and locus overlap/counting.

We apologise for not describing this part more clearly. The term “clumping with GWAS” actually describes the result from the sentence preceding this one. As requested, we have revised this section as follows (lines 395-402):

“To identify lead vQTL with relative independence, we first used LD-clumping⁶⁶ using a p-value threshold of 4.6×10^{-9} , $r^2 < 0.01$ and window size of 5000kb (the same parameter used by¹⁵). The LD analysis between vQTL and lead QTL reported by Vuckovic et al ⁶ used plink 1.9 with the function of --ld. Second, as there are between trait correlations, i.e., blood cell phenotypes correlate with each other, a novel vQTL was defined as follows: 1) was a lead vQTL from the above described clumping analysis, 2) clumped lead vQTL did not have p-value $< 4.6 \times 10^{-9}$ for any blood cell trait levels in Vuckovic et al and 3) was not in strong LD ($r^2 < 0.8$) with reported lead QTL for any trait in Vuckovic et al.”

Reviewer #2 (Remarks to the Author):

This manuscript provides significant contributions to statistical genetics by identifying 176 vQTLs across 29 blood cell traits, with 147 of these not overlapping with conventional additive QTLs. This underscores the importance of variance-based mapping in uncovering previously unexplored genetic loci.

The finding that vPGS can stratify individuals based on genetic variability and enhance the predictive accuracy of PGS is a novel and valuable addition to the field. Demonstrating that combining PGS with vPGS improves trait prediction by ~10% on average is particularly impactful for applications in personalized medicine.

The evidence of stronger negative selection on blood cell trait variance relative to trait levels offers an important evolutionary perspective, providing a nuanced understanding of the selective pressures shaping these phenotypes.

By integrating variance mapping with polygenic scores, this study extends the current understanding of genotype-to-phenotype relationships. The novel use of vPGS addresses an underexplored dimension of phenotypic prediction, advancing both theoretical and applied genetics.

These findings complement and expand prior research on vQTLs in traits like BMI and cardiometabolic biomarkers, but the focus on blood cell traits—ubiquitous in clinical testing—elevates their translational relevance.

i found the following real strengths in reading this manuscript:

The application of Levene's test to identify vQTLs is appropriate for variance mapping, and the use of LDSC to validate inflation factors and assess genetic correlations demonstrates high methodological standards.

Mendelian randomization (MR) analyses to evaluate causal effects of alcohol consumption on trait variance are well-conceived, although the biological plausibility and potential biases (e.g., pleiotropy) warrant additional discussion.

Stratifying individuals using vPGS and testing its impact on PGS predictive accuracy is a compelling demonstration of vPGS utility. However, reliance on PRSice for vPGS construction may limit scalability to non-European populations.

Data Analysis and Interpretation

The results support most of the conclusions, particularly the independence of vQTLs from additive QTLs and the predictive utility of vPGS. However, the interpretation of interactions between PGS and vPGS could be elaborated to clarify their implications for prediction accuracy and biological understanding.

The functional annotations of vQTLs, especially their enrichment in regulatory regions and association with environmental factors like alcohol consumption, are well-documented but could benefit from additional mechanistic insights.

The extensive methods section and availability of summary statistics and code are commendable, making the study highly reproducible. Clear documentation of assumptions (e.g., normality for Levene's test) further strengthens confidence in the analyses.

Thank you for reviewing our manuscript. Please find our responses to the specific comments below.

Following suggestions:

1) While the study highlights the utility of vPGS in stratifying individuals and improving prediction, additional discussion on its potential clinical applications (e.g., identifying high-risk subgroups, informing therapeutic strategies) would strengthen its translational impact.

Thank you for this suggestion. While we are cautious about overinterpreting with respect to clinical applications, we have added the following to the Discussion (lines 303-306): “Nevertheless, we speculate that combining vPGS with PGS could improve genomic prediction performance for those patients at risk. In addition, we identified a list of vQTLs significantly enriched in genomic and epigenomic regulations (Supplementary Data 1), highlighting genes which may be useful for future research on therapeutic targets.”

Clarify Implications of Interactions:

The observed interactions between PGS and vPGS merit further discussion. Are these interactions primarily of methodological interest, or do they reveal biological phenomena that could refine risk prediction models?

It is an interesting question. We have added more discussion regarding the interactions between PGS and vPGS (lines 297-303): “This implies that the effects of PGS on the phenotype can depend on vPGS, which suggests that the non-additive and GxE components related to PGSs could impact prediction accuracy. These findings are consistent with previous observations^{54,55} and may be important for PGS translation. However, to be clear, our observation of the interactions between PGS and vPGS is purely statistical. Future research integrating further molecular data in observational or experimental settings may refine our understanding of these interactions.”

(2) I think you could do a better job of discussing Limitations in Causal Inference. The MR findings linking alcohol consumption to trait variance are intriguing but should be interpreted cautiously. Pleiotropy and potential confounders in MR analyses could be further addressed to contextualize these results.

We agree with this and the above reviewer comment and have added more discussion and interpretative caution in lines 311-319: “In the Mendelian Randomisation analysis, we have chosen alcohol consumption as the exposure as established evidence supports the adverse effects of alcohol drinking on blood cell morphologies^{56,58,59}, likely due to mediation by inflammation and immune responses⁶⁰. To ensure correction of potential pleiotropic confounders we used GSMR as a discovery tool for our Mendelian Randomisation analyses and verification with MR-PRESSO and weighted median. While GSMR and MR-PRESSO both correct for pleiotropic confounders, there could exist other confounders not accounted for in the current study, thus we caution that this is evidence for, but not proof of, a causal interpretation of the effects of alcohol consumption on blood cell trait variances.”

3) The study's focus on individuals of European ancestry is a recognized limitation. Consider discussing how the methodology could be extended or adapted to diverse populations, given the pressing need for equity in polygenic prediction.

We agree and have added the following text in the limitations paragraph (lines 330-336):

“To provide this proof-of-concept, our study was also limited to only European ancestries. However, the challenge of transferability of genetic signals and PGS across ancestries is of high importance and much further research in diverse human populations with paired genomic and blood cell trait data is necessary. Such research should initially focus on multi-ancestry vQTL mapping, combining single-ancestry vQTL mapping results using sophisticated meta-analysis methods, and extend to the latest polygenic score approaches that prioritise ancestry transferability, such as PRS-CSx.⁶³”

4) The study identifies stronger negative selection on variance than on mean levels, which is novel. Expanding on how these findings align with or differ from established theories of stabilizing selection could enrich the evolutionary narrative.

To expand on this, we have revised the following text to the Discussion (lines 267-272), which better contextualises the subsequent discussion of stabilizing selection of neutrophils and inflammation:

“Evolutionary theories show that stabilizing selection will reduce phenotypic variations to maintain population fitness^{13,39-42}. Our results are in line with these theories, although we caution not to overinterpret with respect to the magnitude of negative selection. However, the highly significant negative selection of blood cell trait variances suggests that extreme blood cell levels and morphologies (some of which may be indicative of disease) have not generally been favoured.”